

# Transformation Processes in the Oder Lagoon as seen from a Model Perspective

Thomas Neumann[1], Gerald Schernewski[1], and René Friedland[1]

[1]Leibniz Institute for Baltic Sea Research Warnemünde, 18119 Rostock, Germany

**Correspondence:** Thomas Neumann (thomas.neumann@io-warnemuende.de)

**Abstract.** The Oder/Szczecin Lagoon is one of the largest Baltic Sea lagoons and is subject to very high nutrient loads from the Oder/Odra River. For our study, we employ a modified, high-resolution 3D ecosystem model specifically adapted for this shallow lagoon. The model demonstrates stable and reliable performance over 25 years of simulation, enabling a detailed assessment of lagoon processes under various scenarios. Our model simulations indicate that changes in riverine nutrient

inputs have an immediate impact on the lagoon's water quality, affecting parameters such as phytoplankton biomass and water transparency.

Hypoxia is a prevalent phenomenon, affecting most parts of the lagoon. On average, the lagoon retains 12% (253 t/year) of the phosphorus and 40% (17,278 t/year) of the nitrogen riverine inputs. The primary sink process for phosphorus is sediment burial, and for nitrogen, it is denitrification. Nitrogen retention decreases with increasing riverine loads, dropping to around

30% during years with exceptionally high inputs. The nutrient retention capacity of the lagoon has significant implications for Baltic Sea eutrophication but is not currently accounted for in major policies and Baltic Sea models.

Although recent nutrient loads from the Oder River comply with policy targets, such as the Baltic Sea Action Plan's maximum allowable inputs and Germany's river targets, these levels are insufficient to improve the lagoon's ecological state sufficiently. The Oder Lagoon remains in a highly eutrophic condition, making the achievement of a good ecological status

unlikely.

## 1 Introduction

Marine ecosystems around the world suffer from increasing oxygen deficiencies (Diaz and Rosenberg, 2008). Especially coastal zones are affected by permanent or seasonal low oxic conditions (Fennel and Testa, 2019; Conley et al., 2011; Carstensen et al., 2014). The drivers of deoxygenation are land and airborne nutrient depositions and increased temperature (Kabel et al., 2012;

Börgel et al., 2023). Coastal features such as lagoons and bays can significantly reduce terrestrial nutrient loads. This is due to their extended residence time and shallow waters. Shallow waters facilitate a close interaction between sedimentary processes and those in the euphotic zone, thereby speeding up biogeochemical cycles (e.g. Asmala et al., 2017).

The Baltic Sea includes a diverse array of coastal waters, bays, lagoons, and estuaries, each with unique features and behaviors. These waters host specialized flora and fauna and act as transformers and retention units for external nutrient loads.

Systems with prolonged water residence times and high nutrient loads are particularly important for Baltic Sea pollution. Only



a few coastal waters can significantly alter nutrient loads transported from rivers to the Baltic Sea. Notable examples include the Curonian and Vistula Lagoons, the Gulf of Riga, coastal waters near St. Petersburg, and some Scandinavian estuaries. Among these, the Oder Lagoon is likely the most critical system in terms of quantitative nutrient retention and transformation.

Asmala et al. (2017) studied the coastal filter for the entire Baltic Sea by observations of denitrification and sediment cores.
Their results suggest a removal of 16% nitrogen and 53% of phosphorus from the land based loads. Swedish coastal waters exhibit a relatively high nutrient retention capability. Edman et al. (2018) reported a mean phosphorus retention of 69% and a mean nitrogen retention of 53% estimated from model simulations. These high values, particularly for phosphorus, are due to the oxic water conditions, which favor the trapping of phosphorus in the sediment.

The shallow Oder Lagoon, located at the German/Polish border in the southern Baltic Sea, is one of the largest lagoons in
Europe. With an average water discharge of about 500 $m^3$/s and a drainage area of about 120,000 $km^2$, the Oder River is one of the most important rivers in the Baltic Sea catchment. Due to similarities with other lagoons, it can be expected that insights into retention and transformation processes can be related to driving factors, and simplified relationships can be transferred to other systems. By studying the most important coastal water systems in detail, load calculations to the Baltic Sea can be systematically improved, enhancing regional ecosystem state and water quality assessments.

The European Water Framework Directive (WFD) is the primary legislation aimed at achieving good ecological status in European coastal and transitional waters. Under the WFD, these systems are classified as distinct water bodies, which undergo regular monitoring. Consequently, long-term data on Baltic coastal waters are available, with some records dating back to the 1970s for most countries. However, these data are typically collected fortnightly from a single station intended to represent the entire system. While this is adequate for assessing averaged overall states and long-term changes, the limited
spatial and temporal resolution hinders the analysis and understanding of major processes within these systems and their annual dynamics. Increasing the frequency of sampling and the number of locations would raise costs beyond feasible levels. Therefore, combining 3D ecosystem models with field data is necessary.

Additionally, the Baltic Sea Action Plan (BSAP) (HELCOM, 2021a) establishes the maximum allowable inputs (MAI) for nitrogen and phosphorus required to achieve a good environmental status in the Baltic Sea. These inputs describe loads to the
open Baltic Sea and do not consider possible modifications within the coastal filter, particularly lagoons where large rivers enter.

Recent high-resolution 3D ecosystem models of the Baltic Sea, such as presented by Piehl et al. (2023), effectively and sufficiently describe processes for practical purposes like policy implementation and state assessments. However, these models fall short in semi-enclosed, enclosed, and several transitional waters due to inadequate spatial resolution. Currently, neither data
nor models adequately account for retention and transformation processes in these systems, leading to overestimated nutrient loads to the Baltic Sea. This, in turn, results in inaccurate policy settings, such as MAI or water quality thresholds and targets.

Our objectives are to:

(a)  set up a spatially high resolved ecosystem model for the Oder Lagoon;

(b)  apply and validate the 3D ecosystem model in the Oder Lagoon using long-term data;

(c)  assess the role of spatial model resolution on its performance;



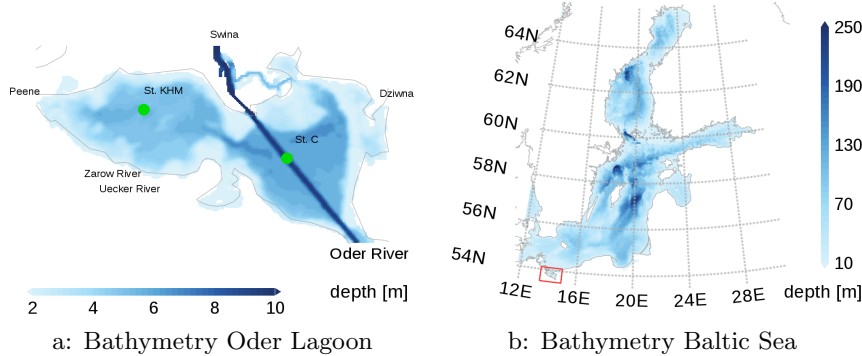

**Figure 1.** Bathymetric map of the Oder Lagoon model (a). The green dots indicate two stations used for validation. Peene, Świna, and Dziwna mark the locations of the open boundary conditions, and Oder, Uecker, and Zarow Rivers represent the river mouths. The red rectangle in (b) shows the location of the Oder Lagoon within the Baltic Sea. The map was created using the software package GrADS 2.1.1.b0 (http://cola.gmu.edu/grads/, last access: 28 November 2024), using published bathymetry data (Seifert et al., 2008).

   (d)  quantify the retention of nitrogen and phosphorus and their inter-annual variability in the lagoon;

   (e)  analyze the driving parameters; and

   (f)  assess the consequences for policy implementation, namely water quality thresholds, acceptable riverine loads, as well as lagoon and Baltic Sea management.

## 65  2  Model Setup

We employ a numerical modeling technique to evaluate the Oder Lagoon ecosystem. This model integrates both biogeochemical and circulation components. The circulation aspect utilizes the Modular Ocean Model MOM5.1 (Griffies, 2004), while the biogeochemical part is based on ERGOM version 1.2 (Leibniz Institute for Baltic Sea Research, 2015). A detailed explanation and validation of the model can be found in (Neumann et al., 2022). For this study, the model was specifically set up for the

Oder Lagoon area within the Baltic Sea. Figure 1 illustrates the bathymetry of the model. A notable aspect is the navigation channel, which has a depth of 10 m. Beyond the channel, the lagoon remains relatively shallow with depths predominantly under 5 m. The horizontal grid resolution is 150 m. Vertically, the model is divided into 28 layers, starting with a layer thickness of 25 cm at the top and 50 cm at the bottom. Three open boundary conditions, Peene, Swina, and Dziwna in Fig. 1, link the model to the Baltic Sea. Data from a coarser Baltic Sea model with a 2 km resolution (Piehl et al., 2023) are applied at the

OBC locations. The Oder River enters into the lagoon from the southern edge.

    The ERGOM model describes cycles of nitrogen, phosphorus, carbon, oxygen, and partially sulfur. Primary production is driven by photosynthetically active radiation, facilitated by four functional groups of phytoplankton (large cells, small cells, limnic phytoplankton, and cyanobacteria). The optical sub-model estimates the light climate based on chlorophyll and CDOM (Colored Dissolved Organic Matter) concentrations (Neumann et al., 2021). Dead organic matter accumulates in the detritus





state variable. Bulk zooplankton grazes on phytoplankton and represents the highest trophic level considered in the model. Particulate organic matter (POC: phytoplankton, detritus, and other POC species) have the capability to sink into the water column and accumulate within a sediment layer. Detritus undergoes mineralization into dissolved inorganic nitrogen and phosphorus both in the water column and in the sediment. The mineralization process is influenced by water temperature and oxygen concentration. In oxygen-rich conditions, phosphate becomes bound to iron oxide and is subsequently retained as particles within the sediments. Erosion resuspends these particles, which are then transported by currents to deposition areas. In anoxic conditions, iron oxide is reduced, releasing phosphate into the water as dissolved phosphate (Neumann and Schernewski, 2008). Oxygen is produced through primary production and consumed by processes such as metabolism and mineralization. Furthermore, the extracellular excretion of dissolved organic matter by phytoplankton results in non-Redfield carbon uptake.

In addition to the model proposed by Neumann et al. (2022), we have introduced a fourth phytoplankton group (limnic phytoplankton). This new group is specifically designed for low salinity and turbid coastal waters, realizing growth limitations due to high salinity levels as well as increased light sensitivity.

All organic particles (phytoplankton, detritus, etc.) are counted in nitrogen units. To compare the model phytoplankton with chlorophyll observations, we sum up all phytoplankton groups and multiply them by a constant chlorophyll-to-carbon ratio.

At the open boundaries, the model is forced by data from a coarse grained model as noted above. Meteorological forcing data are from the coastDat-3 dataset (Geyer and Rockel, 2013). Nutrient loads and runoff into the lagoon from rivers Oder, Ücker, and Zarow were provided by Polish and German national agencies (see *code and data availability*). We start the model simulations in 1995 initialized with data from Piehl et al. (2023) and run it until 2019. For analysis of the simulations, we diagnosed two-day means of all relevant state variables, processes, and transports.

For the analysis of our results, we derived diagnostic variables from the model state variables:

(a) Secchi depth as a function of water constituents: Neumann et al. (2015).

(b) Chlorohyll: The sum of phytoplankton model variables multiplied with a constant chlorophyll to carbon mass ratio of 40 (e.g., Neumann et al., 2015).

## 3 Results

First, we will assess the model's performance, followed by presenting the results from its runs.

### 3.1 Model Skill Evaluation

We will compare observations from two stations, KHM and C (see Fig. 1), with the data simulated by the model. In the figures, model data are shown by solid blue lines and observations by red diamonds. Model data are daily means. Observations are available on request from national agencies (see *data availability*).





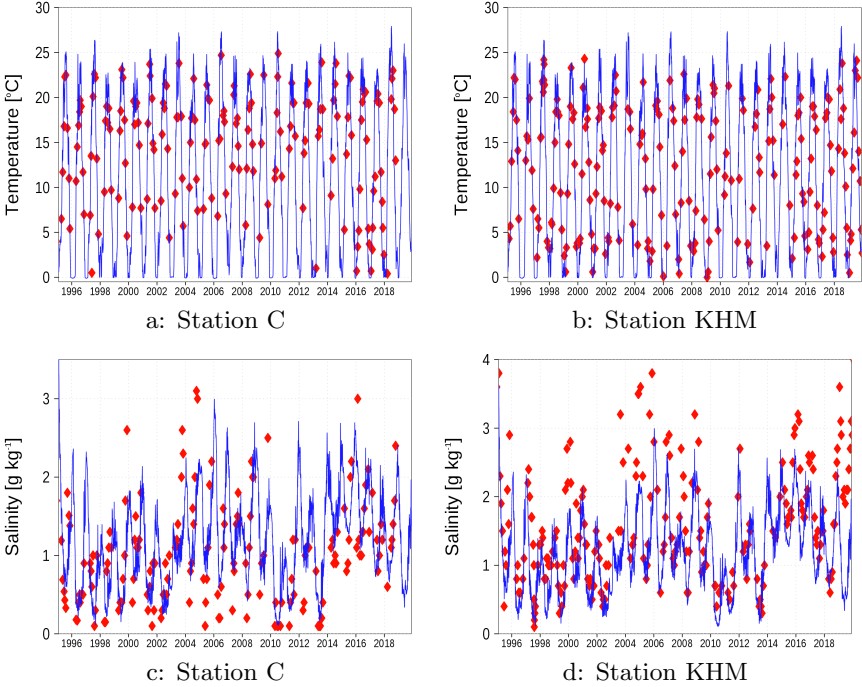

**Figure 2.** Surface temperature and salinity at station C (a,c) and station KHM (b,d) (see Fig. 1). Red diamonds: Observations; Blue line: Modelsimulation.

### 3.1.1 Physical parameters

Figure 2 presents surface salinities at stations C (c) and KHM (d). Simulations at both stations successfully capture the observed interannual variability, although the model fails to reproduce the low salinities observed between 2004 and 2008, possibly due to an underestimation of runoff in the forcing. Corresponding sea surface temperatures (SST) are shown in Figures 2a and b. Winter observations are often unavailable due to sea ice coverage, while in summer, the model tends to exhibit higher SST values than observations, which might not capture peak temperatures. However, this discrepancy has yet to be confirmed.

Water and matter exchange between the lagoon and the Baltic Sea occurs through three channels: the Peene Stream, the Świna, and the Dziwna (Figure 1). The relative contribution of each channel to the water exchange with the Baltic Sea has been estimated in several studies using observations and models. Mohrholz and Lass (1999) provide an overview of known estimates, with the following ranges: Peene Stream: 14%—20%, Świna: 60%—75%, and Dziwna: 9%—20%. Figure 3 illustrates the contributions of the three channels to water exchange in the model simulation. The transport through the Dziwna channel

appears to be higher than in other estimates, whereas the transport through the Świna channel is reduced.





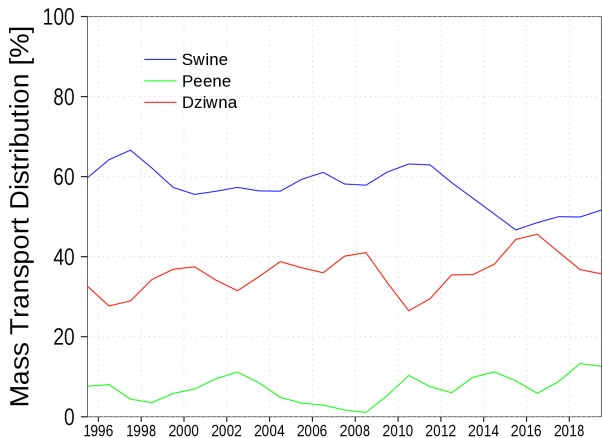

**Figure 3.** The relative contribution of the three outlet channels of the Oder Lagoon to the net water exchange with the Baltic Sea.

### 3.1.2 Biogeochemical parameters

Time series of nutrient concentrations, including phosphate, dissolved inorganic nitrogen (DIN), and chlorophyll, at stations C and KHM are presented in Figure 4. The model successfully captures the decreasing trend in winter phosphate concentrations at both stations. However, it fails to reproduce the exceptionally high observed concentrations in summer. These peak values

are assumed to result from very low oxygen conditions at the sediment surface, which liberate iron-bound phosphate from the sediment. Although this process is included in the model, the amount of phosphate released is insufficient to significantly increase the surface concentration. In certain periods and regions, the model indicates oxygen depletion (see Chapter 3.2), suggesting that the preconditions for phosphate release are met.

The chlorophyll concentration at station KHM is underestimated. Here, we refer to the simple chlorophyll estimation using

a constant Chl:C mass ratio (see Chapter 2). This approach neglects the annual dynamics of the Chl:C ratio, which is a phytoplankton response to changing ambient light conditions and may vary between 23 and 60 (Jakobsen and Markager, 2016).

An extended analysis of the model performance is provided in Appendix A.

### 3.2 Oxygen dynamics

Due to the shallow bathymetry, the water column is well-mixed during winter. However, in summer, stratification may occur, and elevated temperatures accelerate metabolic processes, potentially leading to anoxic conditions in bottom waters. Figure 5a illustrates the total number of oxygen-depleted days, while Figure. 5b shows the average duration of these anoxic events. A notable region of anoxia is found in the navigation channel, which is 10 meters deep – approximately 5 meters deeper than the surrounding area. This deeper channel acts as a sediment trap, increasing oxygen demand for sediment respiration.



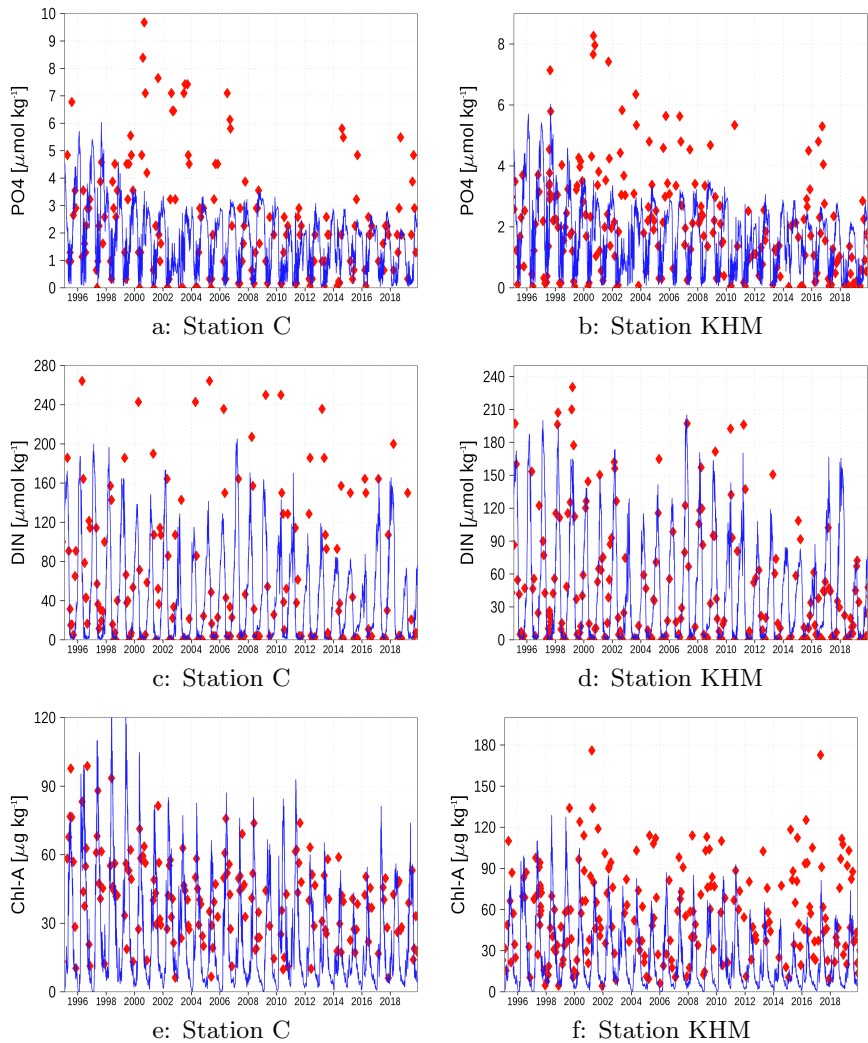

**Figure 4.** Surface concentration of phosphate, DIN, and chlorophyll at station C (a,c,e) and station KHM (b,d,f) (see Fig. 1). Red diamonds: Observations; Blue line: Modelsimulation.

Additionally, ventilation from the surface occurs less frequently here compared to other regions due to the greater water depth. Aside from the navigation channel, the eastern part of the lagoon is more affected by anoxia than the western part. This is likely due to the high nutrient loads from the Oder River entering the eastern lagoon.





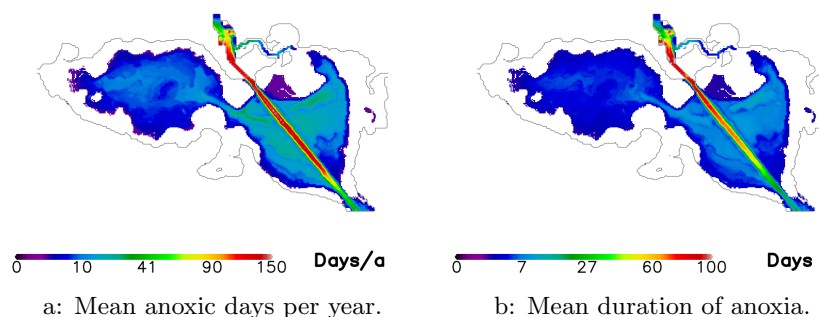

a: Mean anoxic days per year.  b: Mean duration of anoxia.

**Figure 5.** The mean number of days of anoxia in the bottom layer (a) and the mean duration of anoxia (b). The map was created using the software package GrADS 2.1.1.b0 (http://cola.gmu.edu/grads/, last access: 28 November 2024), using published bathymetry data (Seifert et al., 2008).

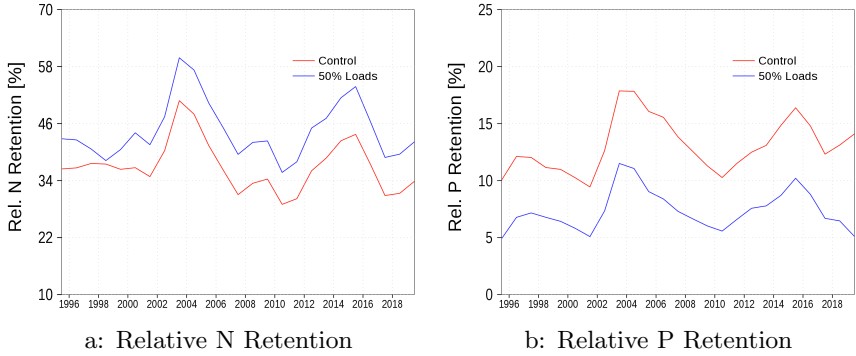

a: Relative N Retention  b: Relative P Retention

**Figure 6.** The relative retention capacity of the Oder lagoon for nitrogen (a) and phosphors (b). The blue line is the sensitivity simulation with halved loads.

## 3.3 Filter function

Lagoons play an important ecological role, particularly through their filtering function, which is crucial for nutrient manage-
ment. Biogeochemical processes within the lagoon help retain or remove nutrients from the system. Consequently, the amount
of nutrients entering the open sea is reduced compared to the amount entering the lagoon.

In this section, we relate nutrient sources to their sinks to determine the proportion of nutrient loads removed within the
lagoon. Additionally, the retention capacity of the Oder Lagoon may be sensitive to variations in nutrient loads. To evaluate
this sensitivity, we performed an additional simulation with a 50% reduction in nutrient input, allowing us to assess how such
changes influence the lagoon's filtering capacity. The results are presented in Fig. 6.



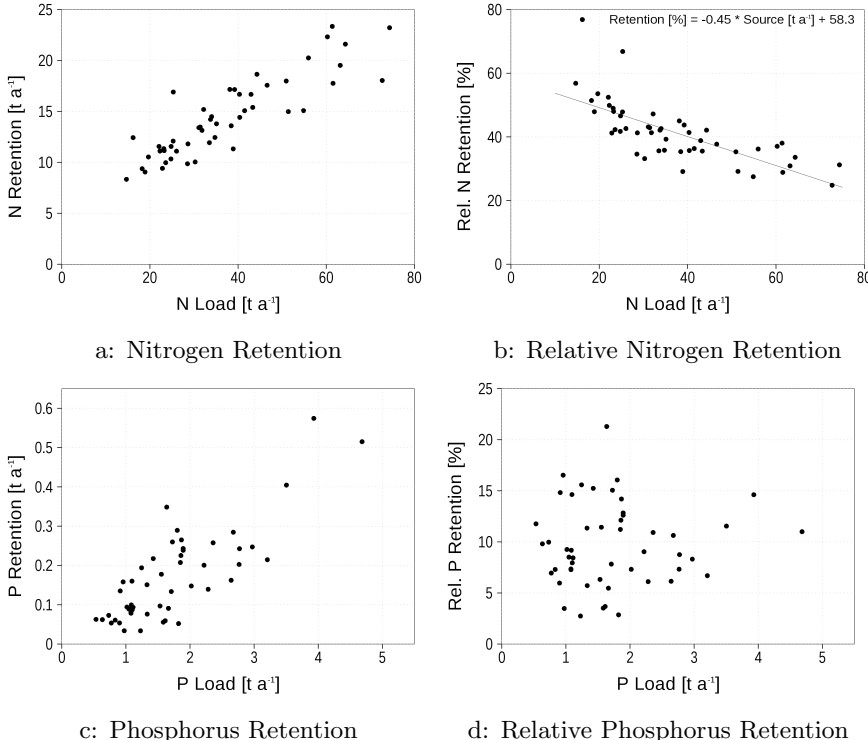

**Figure 7.** Annual means of nitrogen and phosphorus retention (a, c) and relative retention (b, d) in dependence on the loads.

Sources of nitrogen and phosphorus primarily include riverine loads and atmospheric deposition, with riverine loads being the dominant contributor. An additional source of nitrogen is nitrogen fixation by cyanobacteria. Phosphorus has a single sink in the system – burial in the sediment. In contrast, nitrogen has an additional sink through denitrification, occurring both in the water column and the sediment. These factors help explain the contrasting responses of phosphorus and nitrogen retention

capacities to changes in nutrient loads. For phosphorus, reduced loads lead to a decrease in primary production, which results in less organic matter reaching the sediment for burial. For nitrogen, burial also decreases; however, denitrification in the sediment increases because of the greater availability of oxygen.

In the next step, we are testing whether there exists a robust relationship between the retention capacity and the loads for nitrogen and phosphorus. For this purpose, we combined data from the reference run and the run with halved loads to increase

the range of loads and consider the annual means. Figure 7a,c shows the absolute numbers of loads (source) into the system and the retention (sink). The retention increases with the loads, but is this retention at the same rate for all load realizations? Figure 6 suggests that the rate changes with the loads. A detailed dependence of the retention rate on loads is given in Figure 7b,d. In the case of nitrogen, the retention rate significantly decreases with increasing loads. In contrast, a significant relationship between the retention rate and phosphorus loads does not exist, as Figure 6b may suggest.



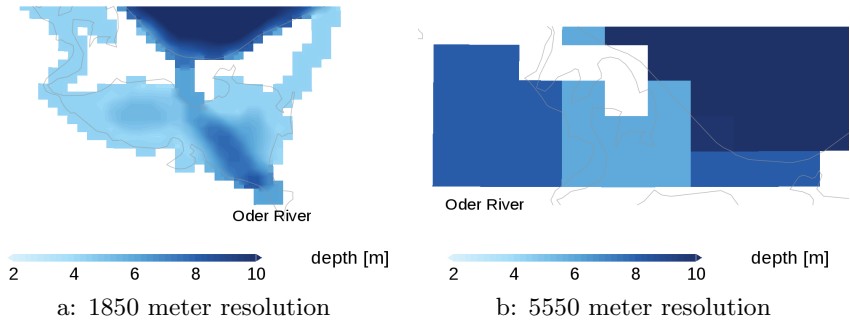

**Figure 8.** Oder Lagoon bathymetry in 1850 meter resolution (a) and 5550 meter resolution (b). The map was created using the software package GrADS 2.1.1.b0 (http://cola.gmu.edu/grads/, last access: 28 November 2024), using published bathymetry data (Seifert et al., 2008).

**Table 1.** Retention capacity for N and P, and for model resolutions 5550m, 1850m and 150m.

|   | 5550m | 1850m | 150m |
|---|---|---|---|
| **N** | 53% | 39% | 37% |
| **P** | 0.7% | 16% | 13% |

In summary, the nitrogen retention capacity is stronger than that of phosphorus, primarily due to denitrification. Approximately 40% of nitrogen and 10% of phosphorus are retained in the lagoon. Reduced nutrient loads to the lagoon increase the nitrogen retention capacity, while the phosphorus retention capacity remains largely independent of load variations. It is important to note that this statement is valid only for the load range applied in the simulations.

### 3.4 Impact of model resolution

In this section, we analyze the impact of the spatial model grid resolution on the simulated retention capacity in the Oder Lagoon. In addition to the 150 meter resolution used in this study, we present results from models with 1850 meter and 5550 meter resolutions. Both coarser-grained models cover the entire Baltic Sea. Figure 8 shows how the Oder Lagoon is represented in the coarser-grained models. While the 1850-meter resolution (Fig. 8a) allows for a more realistic bathymetry, the 5550-meter resolution (Fig. 8b) cannot adequately represent the lagoon. In this coarsest resolution model, we have maintained the lagoon area close to reality.

The average retention capacities for the three different model resolutions are presented in Table 1. While the retention capacities for the 150 meter and 1850 meter resolutions are comparable, the coarsest resolution model performs poorly. It shows almost no phosphorus retention and overestimates nitrogen retention.





## 4 Discussion

### 4.1 Quality of the model results

The Oder Lagoon model demonstrates strong performance in simulating physical and biogeochemical processes. The assessment was conducted at two central stations located in the lagoon's subregions Kleines Haff and Großes Haff, where the best coverage of observations was available.

Temperature and salinity are well reproduced at both sites. However, at station C, the model fails to capture the low salinities observed during the period 2004–2006. A possible explanation for this discrepancy could be the absence of some strong runoff values in the input data during this time. Nevertheless, the model successfully reproduces the interannual salinity variations.

Transport estimates through the connecting channels (Mohrholz and Lass, 1999) suggest that mass transport in the Peene and Dziwna channels is equal, or possibly even lower in the Dziwna channel. However, our simulation yields a mass transport in the Dziwna channel that is twice that of the Peene channel. One possible explanation for this discrepancy could be the incomplete representation of the Dziwna channel in our lagoon bathymetry (see Fig. 8). The absence of the Dziwna channel's full hydraulic resistance in the model could account for the enhanced mass transport observed in the simulation.

A notable characteristic of the simulated nutrient concentrations is the absence of extreme high values, particularly for phosphate. It is hypothesized that these high values are caused by the release of sedimentary phosphate under low-oxygen conditions. Although this process is included in the model, the amount of phosphate released is evidently insufficient to elevate the surface concentration to observed levels. Additional model experiments with increased phosphorus loads in the sediment did not result in surface concentrations approaching observed levels. This discrepancy could be attributed to the two-dimensional sediment module used in this study. Implementing a more sophisticated, vertically resolved sediment module (Radtke et al., 2019) could potentially improve the model's performance.

Oxygen deficiency in the near-bottom water is widespread in the Oder Lagoon. The most affected area is the artificial navigation channel. Being the deepest part of the lagoon, it acts as an accumulation area for sediment and especially for organic matter. Degradation processes consume near-bottom oxygen, and the depth of about ten meters hinders ventilation from the top water layers. Besides the navigation channel, the Großes Haff experiences anoxia more frequently and for longer periods. This is caused by the high nutrient loads from the Oder River entering the Großes Haff. Nevertheless, the model does not account for ship traffic, which causes regular vertical mixing down to the bottom. Thus, in contrast to the model, observations do not show hypoxia in the navigation channel.

The newly introduced phytoplankton functional group (limnic phytoplankton) is by far the most abundant model phytoplankton group. This new group was necessary to achieve a realistic biomass concentration. In combination with the other three groups, we ensure that the biogeochemical model ERGOM can be applied in coastal waters as well as in the open sea without parameter tuning. This is especially important when the model is set up for the entire Baltic Sea at a high spatial resolution, e.g., one nautical mile. In this case, the ecosystem model provides reasonable results in both coastal waters and lagoons, as well as in the open Baltic Sea.





The spatial resolution of numerical ocean models is essential for the quality of simulation results and influences the computational costs of running the model. In general, a finer spatial resolution provides higher quality simulation results but at the expense of requiring more computational power. We have used three different horizontal resolutions to evaluate the impact of resolution on the filter function of the Oder Lagoon: two coarse-grained models of the entire Baltic Sea (5550 m and 1850 m resolutions, Fig. 8) and the local lagoon model with a 150 m resolution (Fig. 1).

The 1850 m resolution version is still able to reproduce the filter capacity of the local model, while the 5550 m resolution version fails (Tab. 1). The consequence is that the coarse-grained model delivers biased nutrient loads to the open Baltic Sea. A solution for upcoming model simulations with a coarse resolution, which are needed for long-term simulations, is to use corrected riverine nutrient loads. This is advisable for rivers entering a lagoon before the open Baltic Sea. For such a procedure, the retention capacity for the contributing lagoons should be estimated with the aid of a local model. Preferably, the dependence of the retention capacity on the loads should also be estimated.

Further discussion of the model performance is provided in Appendix A.

## 4.2 Filter function for nutrients

An important ecological function of the Oder Lagoon is, *inter alia*, the filtration and retention of nutrients. Acting as a system between the Oder River mouth and the Baltic Sea, it reduces the amount of nutrients entering the open Baltic Sea. Budget calculations based on observations in Lampe (1999) yield a low retention capacity of 2%—5%, mainly caused by dredging of the navigation channel. Grelowski et al. (2000) found that 12%-–29% of total nitrogen and 11%-–27% of total phosphorus is retained in the Oder Lagoon. Asmala et al. (2017) showed that denitrification in lagoons of the Baltic Sea is highest, while phosphorus burial in lagoons is small compared to other coastal systems. Our model based approach yields a retention capacity of 40% for nitrogen and 10% for phosphorus. The lower retention for phosphorus is in line with Asmala et al. (2017). In contrast to Asmala et al. (2017), who only considered denitrification, we considered additional sources and sinks for nitrogen: nitrogen fixation and nitrogen burial. Figure 9 illustrates the nitrogen sources and sinks in our model. The dominant source is riverine nutrient loads, while the primary sink is denitrification in the sediment. Dredging is not included in the model. However, its absence is nearly compensated for by the model's sediment burial process, which occurs when the sediment thickness exceeds a specified threshold.

Models offer the advantage of enabling experiments to be performed on the system. We conducted an experiment with halved riverine nutrient loads. The reduced loads resulted in decreased retention for phosphorus, while the retention for nitrogen increased (see Fig. 6). The cause of the lesser phosphorus retention is the reduced primary production due to lower nutrient concentrations. This, in turn, leads to less sedimentation and hence, less burial. However, this dependence is not statistically significant based on our model simulations. Simulations with a broader range of loads could potentially establish a statistically significant relationship. Nevertheless, the loads we applied are within a reasonable range for realistic scenarios. In the case of nitrogen, the contributing sinks behaved differently (see Fig.10). While burial is reduced, similar to the phosphorus case, the relative sedimentary denitrification increases. Denitrification in the sediment is more effective with higher oxygen concentrations at the sediment-water interface. The higher oxygen concentration results from less primary production due to



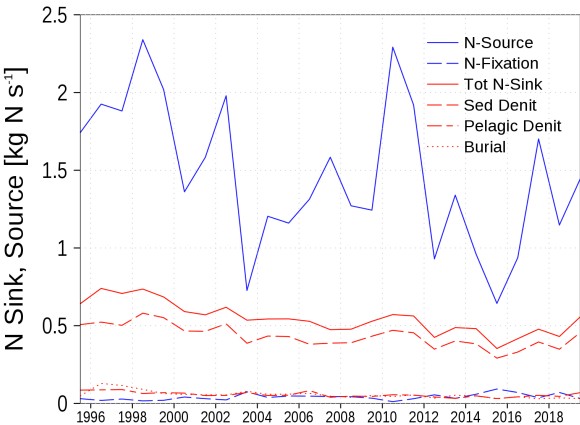

**Figure 9.** Nitrogen sources (blue) and sinks (red) in the Oder Lagoon. N-Sources are riverine loads and atmospheric deposition while N-Fixation is shown as a separate source (blue, dashed line).

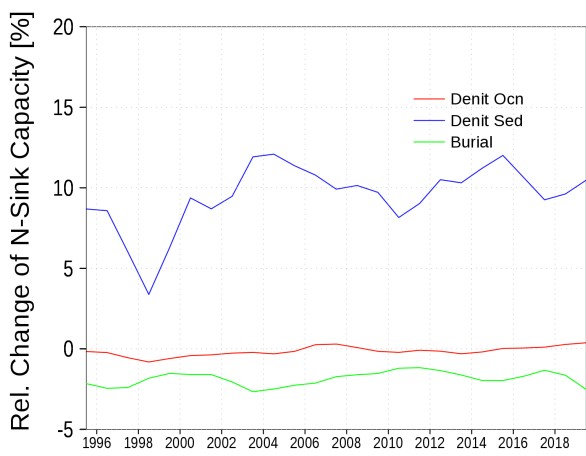

**Figure 10.** Change in relative nitrogen retention capacity due to halved loads.

reduced loads. Overall, nitrogen retention is more effective in the case of lower nutrient loads. A significant correlation could be established for this relationship (Fig. 7).

On average over the 25 simulation years, 40% of the riverine nitrogen (N) loads were retained in the lagoon. Instead of an average total load of 46,266 t/a N entering the lagoon, only 28,988 t/a N entered the Baltic Sea, resulting in a difference of 17,278 t/a N that was retained in the lagoon (Figure 11a,b). With respect to phosphorus, on average over the 25 years, 12% of the riverine phosphorus was retained in the lagoon and buried in sediments. Instead of an average of 2,198 t/a P entering with





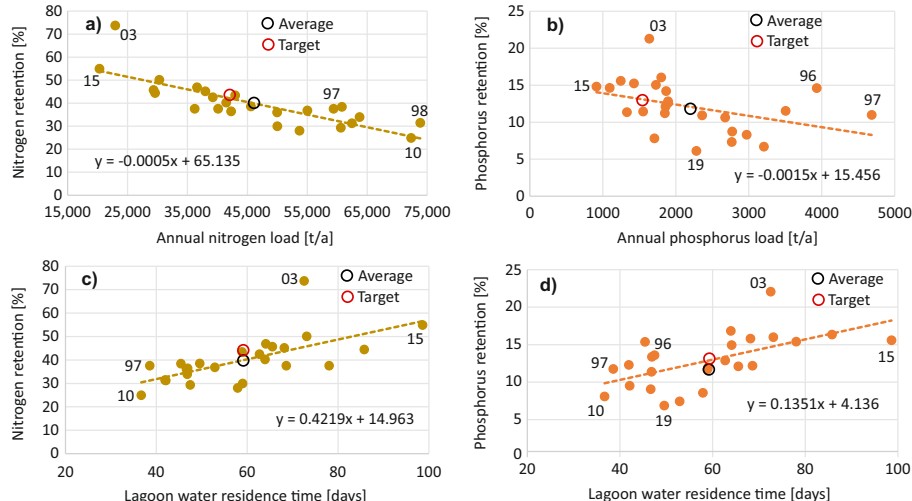

**Figure 11.** Relationship between annual riverine nitrogen and phosphorus loads and the nutrient retention in the Oder lagoon. c,d) Relationship between average annual water residence time and the nitrogen and phosphorus retention in the Oder lagoon.

the rivers, only 1,945 t/a entered the Baltic Sea, resulting in a load reduction of 253 t/a P in the lagoon. Taking the nutrient retention in the lagoon into account, the loads to the Baltic Sea between 1995 and 2019 (28,988 t/a N; 1,296 t/a P) were already far below the BSAP requirements.

The lagoon is an efficient nutrient trap, particularly for nitrogen, and serves as an important purification unit for the Baltic Sea. Especially with respect to nitrogen, a close relationship exists between the relative nitrogen retention in the lagoon and the lagoon's water exchange time. This relationship also exists for phosphorus, but it is less pronounced (Figure 11c,d). In general, we can state that the longer water remains in the lagoon, the higher the relative retention of nutrients in the lagoon.

### 4.3   Annual discharge, nutrient loads and water quality targets

On average, the Oder River contributes 98% of the direct water discharge to the lagoon, while the remaining 2% is contributed by the Zarow and Uecker rivers. The total water discharge into the lagoon, along with the loads of nitrogen (N) and phosphorus (P), exhibits similar temporal behavior (Figure 12a). Between 1995 and 2019, the average water discharge was 518 $\mathrm{m^3/s}$, with average annual total loads of 46,266 t N and 2,198 t P to the lagoon. Between 1995 and 1999, the average water discharge was 643 $\mathrm{m^3/s}$, which is higher than the average over the 25-year model simulation period. Consequently, the annual N loads

(62,534 t N) and P loads (3,600 t P) were also higher. In contrast, during the recent years between 2015 and 2019, the average annual discharge was only 413 $\mathrm{m^3/s}$, resulting in N loads of 37,077 t/a and P loads of only 1,449 t/a. The close relationship between discharge and nutrient loads is illustrated in Figures 12b and c. This relationship emphasizes the dependency of annual riverine nutrient loads to the lagoon on the water discharge of the Oder River. The conclusion is that hydrological processes in the approximately 120,000 $\mathrm{km^2}$ Oder catchment basin exert a stronger control over the nutrient loads to the Oder Lagoon





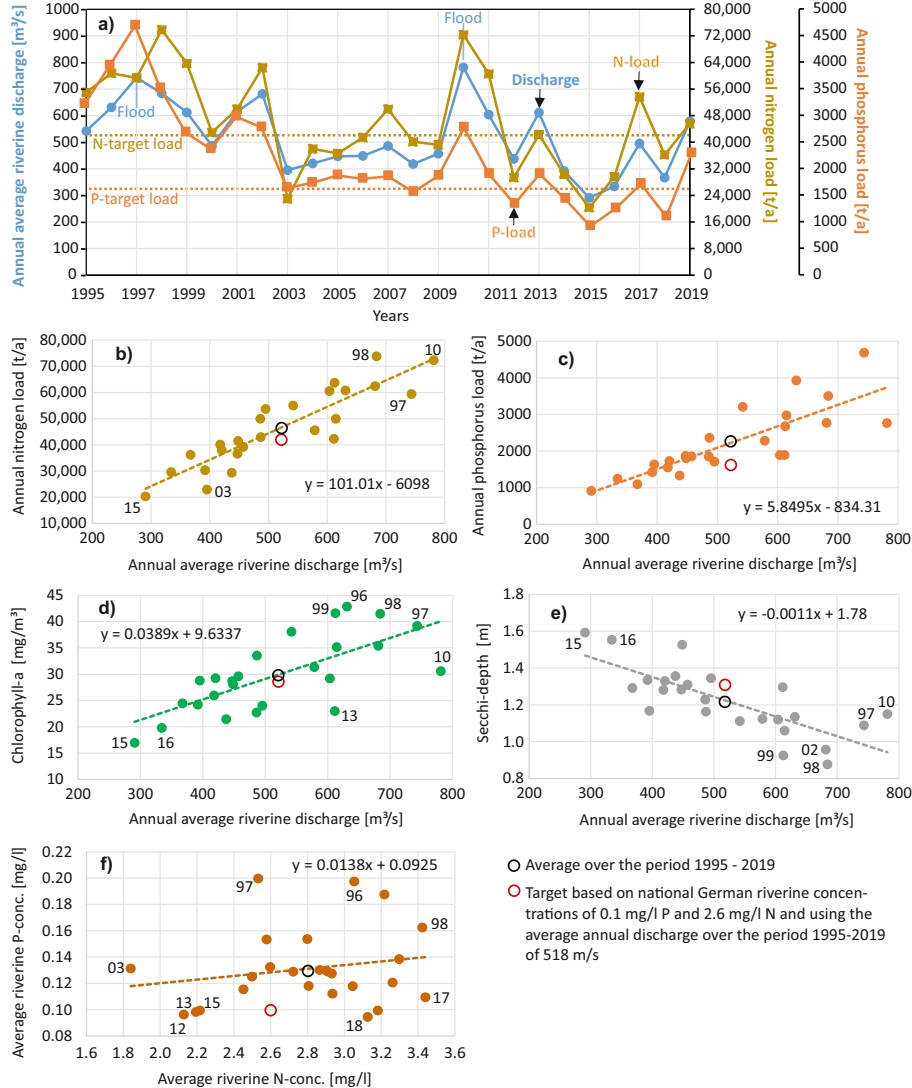

**Figure 12.** Average annual water discharge and annual loads of total nitrogen and total phosphorus to the Oder (Szczecin) Lagoon during a 25 years period (1995–2019), always including the rivers Oder/Odra, Uecker and Zarow. Relationship between riverine water discharge annual riverine nitrogen loads (b), annual riverine phosphorus loads (c), annual average phytoplankton biomass (chlorophyll-concentrations) (d) and annual average water transparency (Secchi-depth) (e) in the Oder Lagoon (spatially integrated). (f) Relationship between riverine phosphorus and nitrogen concentrations.

than annual changes in nutrient emissions. Consequently, future climate change effects on the catchment water budget will significantly impact the riverine nutrient loads and the ecological state of the lagoon.

In Germany, target or threshold values for a good ecological status in rivers exist, with concentrations below these values indicating a good status. The riverine target values are 2.6 mg/L for nitrogen (N) and 0.1 mg/L for phosphorus (P). The nutrient





concentrations in recent years (2015-2019) were already close to these target concentrations. Using the average riverine water

discharge to the lagoon of 518 $\mathrm{m}^3/\mathrm{s}$ (1995–2019) and the German nutrient target concentrations, the resulting target loads would be 42,506 t/a for N and 1,635 t/a for P. The HELCOM Baltic Sea Action Plan (BSAP) defines the maximum allowable nutrient loads that enable the Baltic Sea to reach a good status. According to the BSAP, the maximum allowable loads are approximately 45,000 t/a for N and 1,500 t/a for P. Despite following different approaches, the German target values in rivers and the HELCOM maximum allowable loads result in comparable targets for the Oder River. In contrast, the Polish targets

would allow much higher Oder River nutrient loads to the Oder Lagoon (63,661 t/a for N and 4,615 t/a for P), as shown by Friedland et al. (2019). However, during recent years (2015–2019), the N loads to the lagoon were 37,077 t/a and the P loads were 1,449 t/a. For both nutrients, the loads were below the maximum allowable inputs (HELCOM, 2013).

According to our model simulations, riverine nutrient loads have an immediate effect on major water quality indicators, namely phytoplankton biomass (chlorophyll-a) and water transparency (Secchi depth), and show a close correlation (Figure

12d,e). Increased discharge, resulting in increased nutrient loads, causes an increase in chlorophyll-a concentrations in the lagoon and a decrease in Secchi depth. Calculated chlorophyll-a and Secchi depth values are average annual values over the entire lagoon and cannot be directly compared with existing ecological target values for central stations in the lagoon (Schernewski et al., 2015). However, the changes in chlorophyll-a concentrations and Secchi depth that would result from fully meeting the nutrient load targets (42,506 t/a N and 1,635 t/a P) are only minor (Figure 12d,e). Compared to the average values

over the period 1995–2019, chlorophyll-a concentrations would decrease by 5%, and Secchi depth would increase by 8%. Although. the nutrient loads are below the BSAP values, the lagoon remains in a highly eutrophic state with a bad ecological status, according to the official HELCOM HEAT HOLAS 3 (HELCOM, 2021b) eutrophication status assessment (BMU, 2024). Meeting water quality targets in rivers and the Baltic Sea does not guarantee that a good status in transitional ecosystems, such as lagoons, will be achieved. A good ecological status in the lagoon would require significant additional nutrient load reductions

and the implementation of measures in the river catchment at high and likely unrealistic costs. It is likely that the lagoon must be regarded as a naturally eutrophic ecosystem with limited management possibilities. A re-evaluation of water quality targets in the lagoon requires a more detailed study that includes neighboring coastal waters to address interrelationships, relates model data to field data with a focus on the assessment stations, and carefully considers evaluation aspects such as water depth and evaluation period.

**4.4 Interannual variability of discharge and loads and its consequences**

The interannual variability of water discharge and nutrient loads is high (Figure 12a). Annual discharges vary between $291\mathrm{m}^3/\mathrm{s}$ in 2015, a dry and hot year, and $781\mathrm{m}^3/\mathrm{s}$ in 2010, the year with one of the largest Oder River floods ever recorded, which occurred in May. Consequently, nutrient loads also show high variability, ranging from 20,309 t/a N in 2015 to 72,333 t/a N in 2010. For P, the range is from 4,683 t/a in 1997, another major river flood year, to 1,094 t/a in 2018, another hot and dry

year. Over the 25-year period, riverine concentrations of N and P do not show a close relationship; the concentrations of both elements behave differently in different years. The differing interannual variability between N and P concentrations indicates that both nutrients enter via different pathways and that these pathways play different roles in different years. This suggests that





**Table 2.** Loads for different periods and targets into the Oder Lagoon and Baltic Sea. Loads and targets are in kt/a, runoff in $\mathrm{m^3 s^{-1}}$. N and P retention is estimated from the period 1995 to 2019. Loads to the Baltic Sea for other periods are estimated from the mean retention.

| | N-load Lagoon | N-load Baltic | P-load Lagoon | P-load Baltic | Runoff | N-retention | P-retention |
|---|---|---|---|---|---|---|---|
| 1995–2019 | 46,266 | 28,988 | 2,198 | 1945 | 518 | 0.37 | 0.12 |
| 1995–1999 | 62,534 | 39,181 | 3,600 | 3,186 | 643 | | |
| 2015–2019 | 31,077 | 23,231 | 1,449 | 1,282 | 413 | | |
| German target Oder River | 42,506 | | 1,635 | | | | |
| BSAP MAI to Baltic | | 45,000 | | 1,500 | | | |

seasonal discharge and nutrient emission patterns need to be analyzed separately and in depth. It is likely that the interannual variability is controlled by extreme events lasting weeks to a few months, which can strongly affect the annual values and the lagoon ecosystem.

## 5 Conclusions

We developed a local model for the Oder Lagoon that realistically reproduces its physical and biogeochemical properties. This model serves as a tool for conducting studies in this area and, specifically, for quantifying the lagoon's nutrient retention capacity. This is a crucial step in adjusting riverine loads for coarse-grained models, which often do not adequately resolve lagoons. Our approach can be readily applied to other lagoons in the Baltic Sea, such as the Curonian Lagoon, and can also be adapted for regions beyond the Baltic Sea.

The analysis shows that the nutrient retention in the lagoon already reduces nutrient loads to the open Baltic Sea in agreement with the BSAP (Table 2). Furthermore, riverine loads into the Oder Lagoon also meet the German targets for the Oder River. However, nutrient concentrations in the Oder Lagoon do not achieve the intended target values for good ecological status. This points to the problem that quality standards for inner and outer waters are not harmonized, making it realistically impossible to achieve a good ecological status for the Oder Lagoon.

The simulation data generated in this study enable several additional analyses that could be the focus of future studies. For example, these could include short- and long-term responses to extreme events such as floods and droughts. Further improvements should include a more realistic representation of the Dziwna channel length and coupling the model with a more advanced sediment model than the one used in this study. Another weakness is the lack of ventilation of the deep water in the navigation channel due to ship traffic, which should be properly parameterized in an upcoming model version.





*Code and data availability.* Observations from the Oder Lagoon monitoring program were provided upon request by the German State Agency for Environment, Nature Conservation and Geology Mecklenburg-Vorpommern (Mario von Weber, LUNG-MV) and Główny Inspektorat Ochrony Środowiska (Adam Czugała). Nutrient loads and runoff data for the Oder River were obtained from from the Polish
Główny Inspektorat Ochrony Środowiska. Corresponding data for the rivers Zarow and Ücker were available from LUNG-MV.

The meteorological forcing is archived at https://doi.org/10.1594/WDCC/coastDat-2_COSMO-CLM (last access: 26 November 2024, Geyer and Rockel (2013)).

All model output that was analyzed in this paper, the model code, and data to run the model has been published at https://doi.org/10.5281/zenodo.14236528 (Neumann et al., 2024).

## Appendix A: Additional model skills evaluation

In this section, we show additional model skills assessments.

### A1 Horizontal patterns

Simulated data are assessed in relation to monitoring data, allowing the computation of the root mean square deviation (RMSD) for each station and parameter. Following Kärnä et al. (2021), the RMSD is normalized by the standard deviation of the
observations to enable comparability between different stations and parameters. A dimensionless normalized RMSD value below 1 is often regarded as a good indication of model skill, suggesting that the RMSD is below the natural variability of the observations.

Figure A1 shows horizontal surface patterns of model variables and the corresponding RMSD. For the winter season, observations were not available for all stations.
Dissolved inorganic nitrogen (DIN) follows a clear gradient in both seasons, with the highest values near the mouth of the Oder River, while concentrations decrease drastically towards the western lagoon and the open Baltic Sea. The normalized RMSD for the monitoring stations is mostly below one, indicating good model skills. Only at station E (near the mouth of the Oder River) does it get slightly worse.

Dissolved inorganic phosphorus (DIP) shows an opposite gradient during winter, with the highest concentrations in the
western lagoon. During the growing season, DIP concentrations are high. While the normalized RMSD during the growing season is mostly below one, except at station E near the Oder River, the model DIP concentration in winter is higher than the observed values. For the rest of the year, DIP is within the range of the observations (see Fig. A2).

Summer chlorophyll-a concentration and Secchi Depth (Figure A1) show comparable gradients because both are closely linked. The lowest chlorophyll-a concentrations appear in the western part of the lagoon, where the Secchi Depth is deepest.
Although chlorophyll-a is underestimated in the western part, the normalized RMSD is within the range of the eastern stations. For Secchi Depth, the normalized RMSD is even better for stations in the western part (values between 1.1 and 1.2), while it is up to two for eastern stations. Despite the good agreement between modeled and observed chlorophyll-a in the eastern part of the Oder Lagoon, the September decline of chlorophyll-a is too fast at station C, causing a too rapid increase in Secchi Depth in September (see Figure A2).





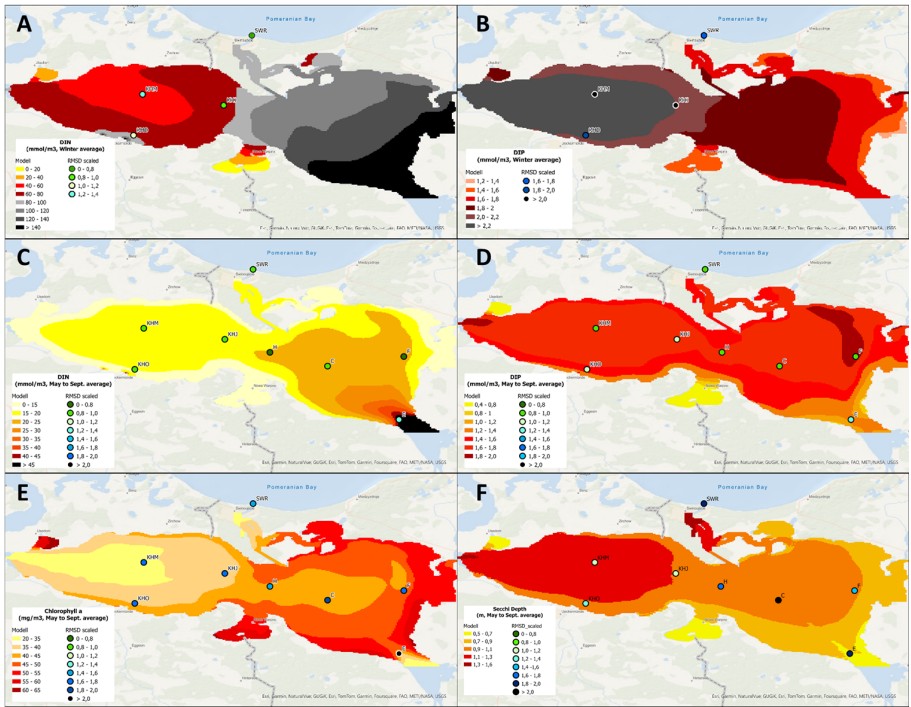

**Figure A1.** Near surface concentrations of dissolved inorganic nitrogen A: Winter average, C: May to September average; dissolved inorganic phosphorus B: Winter average, D: May to September average; Chlorophyll-a E: May to September average, and Secchi Depth F: May to September average. All parameters are averaged from 2010 to 2019. The model results are color-coded and circles indicate the normalized RMSD for each monitoring station and parameter. The map was created using the software ESRI ArcGIS Pro (Version 3.3.2; https://pro. arcgis.com). The background map is the World Ocean Base map published by ArcGIS under the Esri Master License Agreement with contributions from Esri, Garmin, TomTom, GUGiK, Foursquare, NaturalVue, FAO, METI/NASA, and USGS (https://www.arcgis.com/home/ item.html?id=5d85d897aee241f884158aa514954443). .

## 360  A2  Climatologies

In Figure A2, we present the 2010 to 2019 climatology of model data alongside observations (gray dots) at stations C and KHM (for station locations, see Figure 1). In general, the model is able to capture the variability and amplitude of the observations with some exceptions:

(a) The model does not reproduce the extreme high DIP values, which can be attributed to insufficient sedimentary phos-
phorus liberation under low oxygen conditions.

(b) The chlorophyll-a concentration at station KHM is underestimated. This discrepancy can be attributed to the constant carbon to chlorophyll mass ratio used to estimate chlorophyll from the model variables. This approach neglects photo-acclimation of phytoplankton.





**Figure A2.** Monthly climatologies (2010 to 2019) of near surface concentrations of dissolved inorganic nitrogen (A, B), dissolved inorganic phosphorus (C, D), Chlorophyll-a (E, F), and Secchi Depth (G, H) for Station C and KHM.



(c) Secchi Depth is overestimated at station KHM, which is a direct consequence of the underestimated chlorophyll-a con-
centration.

*Author contributions.* TN set up the model and conducted the simulations. All authors contributed to the study design, data analysis, and the writing of the manuscript.

*Competing interests.* The corresponding author has declared that none of the authors have any competing interests.

*Acknowledgements.* The authors gratefully acknowledge the computing time made available to them on the high-performance computer
"Emmy" at the NHR Center NHR-NORD@GÖTTINGEN. This center is jointly supported by the Federal Ministry of Education and Re-
search and the state governments participating in the NHR (www.nhr-verein.de/unsere-partner). We thank LUNG-MV (Mario von Weber)
and Główny Inspektorat Ochrony Środowiska (Adam Czugała) for providing German and Polish monitoring data. This work was financially
supported by the German Federal Ministry of Education and Research, projects "Coastal Futures II" (grant number 03F0980B), "Prime Pre-
vention" (grant number 03F0911B) and UBA-MoSea (FKZ 3723252040). We acknowledge the contributions of Sarah Piehl, who collected
and prepared data from German and Polish authorities.





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
