# Peer review of "Transformation Processes in the Oder Lagoon as seen from a Model Perspective"

_EGUsphere, 2024_

## Referee Comment (RC1)

**Review for «Transformation Processes in the Oder Lagoon as seen from a Model Perspective» by Neumann et al.**

**Overall evaluation**

In general, the manuscript is well written, well presented and clear, including mathematical expressions and equations. It reaches innovative and robust conclusions that can greatly improve model studies in the area. Especially as little is known about coastal retention in the Baltic Sea.
It is also relevant for policy measures, because official nutrient reductions do not account for nutrient retention in coastal waters. However, the conclusions could be more specific and, with additional discussions, could be more relevant for the entire Baltic Sea. While their methodology is valid, clear and relevant, it is difficult to be applied by any other institute studying the Baltic Sea's biogeochemistry, as setting a coastal model for one single region is costly (manpower, budget, time) and not all may have that possibility. However, by expanding their analysis further and adding more discussions on how the Order Lagoon compare to other coastal areas in the Baltic Sea, approximations for N and P retention could be estimated for other areas as well. Published results for the Baltic Sea mentioned in the paper have large discrepancies with specific coastal studies, and therefore new suggestions will be relevant to address including approximations for other coastal areas, especially for those similar to the Oder Lagoon. Their conclusions do mention that the method can be adapted for other regions beyond the Baltic Sea, but as written now is too vague. If this means adapting their model to a new area, it could translate as a costly task (time and computational) and not necessarily straightforward. In this sens, a better clarification on what is possible to extrapolate to other areas with this method is required. In other words, the main missing discussion in this manuscript is: Can the relation found in this study between loads and nutrient retention be used to estimate %retention in other basins? What can be used from these results to improve retention estimates in other coastal areas in the Baltic Sea and other Seas?

The Scientific significance of this study is good and after some revisions, will contribute significantly to our understanding of nutrient retention in coastal waters. The latter is poorly known and therefore, the content of this manuscript is very valuable. The methodology is clear and results relevant to biogeochemistry.

The scientific quality of the study is fair, the methods and approach are valid and innovative. However, it misses additional discussions, described in the specific comments below.

The presentation quality is good, the structure and figures are clear and well explained. However, it misses some additional figures, for example, in the validation section time series of model and observed oxygen should be shown, as well as more on the water column dynamics in the model. The oxygen section can also be expanded, as it is a major parameter affecting both N and P. Section 4.4 should be expanded or moved to the results section. For details, see specific comment below.

In short, I would strongly recommend this manuscript for publication after minor/medium revision.

**Specific comments**

Introduction
line 21: What is meant by extended residence time? Long? If so, replace by "long"
line 26: Please expand and add references. Why are there only a few coastal waters that can reduce nutrient loads in the Baltic Sea? Why not all coastal areas have that capacity? The Baltic Sea has many different types of coastal waters, please mention them as well and why would they be less

effective in nutrient retention.

Line 28: Why is the Order Lagoon the most critical?

Line 48: Are the MAI here referred per basin in the Baltic Sea or to the entire Baltic Sea?

Line 55: Models can account for nutrient retention in the Baltic Sea taking values from literature and making assumptions for unstudied coastal areas. They can give good enough nutrient concentration near coastal areas, but can certainly be improved. The problem is not that they do not adequately account for it, but that there is not enough information to adequately account for all different types of coastal areas of the Baltic Sea. In this regard, this study can really improve such model results. Here, I would suggest to focus more on the fact that there is missing information on this for the Baltic Sea (and other Seas) and discuss this further. Importantly, this has been already mentioned in previous modeling studies, for example in:

Eiola et al., 2011, they mention that "*the major differences between the nutrient supplies to the different models are due to the different assumptions on the bioavailability of phosphorus loads*", referring to the retention of P in coastal waters. They also have a dedicated chapter on bioavailable nutrient loads in their discussions, which should be mentioned in this study.

Ruvalcaba-Baroni et al., 2024, they refer to the little information available and what can be currently used: "*As the response to nutrient removal of different coastal types is poorly quantified... these factors are taken from previous studies in the Baltic Sea (Eilola et al., 2011; Edman and Anderson, 2014; Asmala et al., 2017).*" In their discussion, they also mention that "*one factor affecting detritus is the fraction of the organic matter coming from rivers that is actually bio-available and not directly retained in coastal waters.*" They assume a constant fraction for the entire Baltic Sea and say that "*the input of organic matter from rivers, especially nitrogen, could be improved by better accounting for river-specific organic matter retention in coastal waters*".

Adding this discussion, will make the point of this study stronger, as it "partly" answers a problem that has been existing since many years.

Results

line 112: The authors mention that "the model fails to reproduce the low salinities observed between 2004 and 2008, possibly due to an underestimation of runoff in the forcing", but do not show the forcing used. What evidence do you have for this? It could also be due to the atmospheric forcing (precipitation vs evaporation). Please add more explanation for this, a supporting figure with, for example the runoff used in the high resolution model and how it compares to observations could be helpful here (or in supplementary). Also, why not show salinity and temperatures profiles or some water column performance?

Line 114: The authors mention that "However, this discrepancy has yet to be confirmed". This statement falls a bit short. How would you confirm this and why not doing it in this manuscript? Please rephrase or add information.

Line 123: The authors say "the model successfully captures the decreasing trend in winter phosphate". Plotting the trend directly in the figure for both model and observations would make this point more clear. It will also be relevant to plot it for the other biogeochemical parameters. The significance of the trend can then be estimated, which, for example, it is not entirely clear for chlorophyll-a at KHM (is the decreasing trend significant or not?). This information would be valuable for the discussions later on and for line 129, where the model and observed chla are described. From Fig 4, it looks like the model has also a more pronounced decreasing trend than the observations at KHM, but very good match at C.

Line 131-132: The authors mention that "phytoplankton response to changing ambient light conditions and (???) may vary between 23 and 60". I think the authors missed writing "Chl:C ratios". Please add the missing part. If so, would the Chl:C ratio be that different between C and

KHM? Would this alone explain the discrepancies in model performance between C and KHM?

Oxygen dynamics (section 3.2)
This section is very relevant to the manuscript. While the model shows oxygen depletion, there is missing evidence on how depleted the oxygen is in the area. Why not add oxygen time series in bottom waters from both observations and model? Is there any oxygen measurements in this area? This requires further discussions either to show more evidence for oxygen depletion or mention the lack of observations and what are the alternatives to "validate" the model in such case. What are the references for oxygen depletion in this area? This is also relevant for the statement that the authors write in line 126 "..., the amount of phosphate released is insufficient to significantly increase the surface concentration." This is linked to oxygen, but also to mixing. Neither are shown. Water column dynamics can be shown in different ways (e.g., salinity and temperature profiles, mixing water depths, etc). It is difficult to judge how well the model represents the oxygen concentration in bottom waters or phosphorous retention in the sediments. It may as well be that the release of phosphorus is not insufficient in the model, but that the lower values in the surface are due to a biased runoff forcing that stratifies the water column too much in the model. Please add water column dynamic information (it can be in the appendix) and evidence for low oxygen concentrations in bottom waters from observations (e.g., time series for C and KHM of bottom oxygen). In figure 5, an additional map showing the actual model oxygen concentrations in bottom waters for the months with oxygen depletion and available observations will be valuable.

Line 164: Phosphorus retention in the sediments is also affected by bottom oxygen concentrations. It will be interesting to plot retention rates vs oxygen concentrations in bottom waters, especially for P. Could this (partly) explain the lack of relationship between rates of retention and P loads?

Line 175: How can the higher resolution model overestimate nitrogen retention?

Discussion
Line 185: The discussion relates to runoff forcing and would benefit from a runoff figure (as mentioned in comments above for line 112).
Line 195: See comment above on Oxygen dynamics (section 3.2). This discussion should be expanded.
198: There is much more literature available for P recycling in the Baltic Sea that can help tuning the model. Please add more references.

Lines 199-205: Relates to comment above on Oxygen Dynamics. This paragraph does not have any reference. How do we know the Order Lagoon experiences anoxia? Oxygen is a relevant topic for this type of study. Please expand the discussions on this and evidences for oxygen depletion available for the Order lagoon.
Line 205: So there are observations? Why not show them?

Line 220: Pleas add "As done in Eiola et al., 2011 and Ruvalcaba-Baroni et al., 2024, where they use a factor to account for nutrient retention" or similar (see comment above for line 55).

Line 221-222: Please expand. This has more potential and general suggestions on how to improve model estimates in the Baltic Sea can be given based on the relevant results in this manuscript. The big picture would be to say something on how these very nice results (based on chapter 4.2, 4.3 and 4.4) can be used by the larger scientific community and perhaps even policy. This could be discussed in a chapter on its own (4.5?).

Line 222: As mentioned before, not all coastal areas can be modeled the way it was done in this manuscript (it will be nice, but it will take time). Please expand on how your results could differ in

other areas, based on bathymetry, residence time, oxygen content, etc and how they relate to published values (some of which are already mentioned in the introduction, but there are a few more, see discussions in Eiola et al., 2014). Or is it that every single coastal area in the Baltic Sea needs to be modeled separately to be able to estimate their coastal nutrient retention? Perhaps, but some relationships must be similar.

Line 240: There is no statistics shown in the paper, how do you know it is "not statistically significant"?

Line 258: This is an important statement that could be repeated in the conclusion.

Line 291: Please replace "." by ","

Interannual variability of discharge and loads and its consequences
This entire section is structured more as results than as discussions. Please expand (add discussion and references) or move to results.

Line 308: Please replace "emissions" by "inputs"

Line 310: How do you know that interannual variability is likely controlled by extreme events? Is there a reference or some evidence? It may seem obvious, but the relative or quantitative impact remains quite unknown for the Baltic Sea, as far as I know. Also what do you mean by "extreme events"? Storms, heatwaves and/or what else?

Line 315: How easy/difficult would it be to expand it to other regions in and outside the Baltic Sea? Please clarify.

320: I am not sure that the word "realistically impossible" can be used in this study, as it does not assess economical impacts and costs. Maybe rephrase to something less strong.

Figures
Figure 1. It would be good to mention already in the caption the navigation channel. It is quite obvious that there is a channel, but is not redundant to mention it. It would be also interesting to know when was this channel made (before the 60s?) to have an idea on how far back in time we should go to get "pristine conditions" in this area and to better interpret the observational data sets in this area as it seems to have a large impact for nutrient retention.

Figure 5. Please add the period mean in the caption.

---

## Referee Comment (RC2)

Comments on "Transformation Processes in the Oder Lagoon as seen from a Model Perspective"
by Neumann et al.

**General comments:**

In the manuscript by Neumann et al., the authors discussed the function of the Oder Lagoon in nutrient retention using a 150-m 3D ecosystem model. The lagoon was found to significantly affect the Baltic Sea eutrophication given the high nutrient retention capacity by the estuary (12% of riverine phosphorus and 40% of riverine nitrogen). However, the lagoon itself was found in a highly eutrophic condition which requires further attention and studies. The manuscript is well presented, clear, and easy to follow. However, as a model paper, the model evaluation section is too weak, lacking assessment of some key feature like, oxygen concentration, chl-a concentration (against satellite), and T/S vertical profiles. In addition, there is a lack of in-depth discussion in nutrient retention. I will list my comments in detail as followed.

Therefore, I recommend a major revision before publication.

**Detailed comments:**

1. Line 27. Please add references for the examples.

2. Line 49. Could you be more specific about what the definition of "a good environmental status" are?

3. Line 61. Why did the authors study the inter-annual variability instead of other scales like seasonality which should be more apparent than inter-annual signal for many earth systems?

4. Section 3.1. As a model paper, I think it is not enough to evaluate surface T, S, nutrient, and chl-a concentration over such big area, especially as the authors pointed out that water stratification was found as a factor to bottom dissolved oxygen deficiency. Evaluations of T/S vertical profiles, nutrient profiles, oxygen profiles, and chl-a spatial patterns (against satellite estimates) are crucial to validating the modeled results.

5. Figure 2. Does the model represent sea ice dynamics? I found that the modeled water temperature in winter is > 0 °C. As the authors mentioned at Line 113, the SST observations were not available in winter due to sea ice coverage. So, in reality, there should be sea ice covered the lagoon in winter. Please clarify the model's capability in sea ice modeling.

6. Figure 2 caption. "Modelsimulation" should be "Model simulation".

7. Lines 110-114. More quantitative comparison is needed like providing $R^2$, RMSE, and relative RMSE.

8. Line 111. As I observed, salinity was overestimated at Station C, but was underestimated at Station KHM between 2004 and 2008. Also, the authors need to provide evidence to support "possibly due to an underestimation of runoff in the forcing". The runoff forcings are from observations. If the statement is true (i.e., runoff is underestimated), it may indicate that the number of river point sources are not enough, or the atmospheric precipitation is underestimated. Please show the evidence to support the causes of underestimation of salinity.

9. Lines 118-120. Why there is such a great difference in transport through the Dziwna between your estimates and others?

10. Figure 3. Is the plot based on monthly average? Please provide the way how the transport contribution is calculated.

11. Lines 125-127. There is a lack of evidence to address (1) that low bottom DO triggers the release of iron-bound phosphate from sediment and (2) that the amount of phosphate released is insufficient to increase the surface concentration.

12. Line 139. I am not convinced that the deeper channel acts as a sediment trap. Instead, I believe water stratification is the primary factor contributing to bottom DO depletion in the main channel. First, surface chl-a concentrations do not show a distinct pattern inside versus outside the main channel (Figure A1e). This suggests that the amount of sinking organic matter should follow a similar spatial distribution, especially in such a shallow lagoon (< 10 m), where organic matter does not drift far from where it sinks. Second, water column stratification plays a crucial role in the development of bottom hypoxia, as demonstrated by numerous hypoxia studies. As the authors later mention, the model does not account for mixing processes due to heavy traffic in the main channel, which leads to discrepancies between the modeled and observed bottom DO concentrations (hypoxia is modeled but not observed in the measurements). This suggests that bottom hypoxia in the main channel may be more influenced by strong (or overestimated) stratification.

13. Figure 4. Statistics like $R^2$ and RMSE should be provided.

14. Lines 151-157. I think this is the core of this work. The authors may need to provide a diagram for quantification of each source and sink terms for both N and P. I found the Figures 9-10 attempt to address it, but there is a lack of quantification for the P sources and sink terms. Also, it is better to move Figure 9-10 to here.

15. Line 157. As I understand, denitrification occurs at anoxic conditions. That is, denitrification rate at sediment should decrease as oxygen concentration increases.

16. Line 160. Why don't you use the daily mean to increase the sample size? As I observed, the sample size in Figure 7 is small which may weaken the conclusion drawn.

17. Figure 7. The plot read confused to me. The authors mix the output of the control and reduced load experiments when generating this plot. However, the plot includes two types of signals: (1) annual signal of the retention rate which changes as nutrient loads; (2) retention rate changes due to the changing system when nutrient is reduced. The former is the one what the authors want to analysis. However, regarding the latter one, when the total nutrient loads are reduced, the entire ecosystem will adapt to such changes and turn out to be a new system. For example, some phytoplankton species can become the dominate species given their higher adaptation to low-nutrient environment. Such changes may affect the sinking organic matter not

just in spatial distribution but also in temporal phases. So, my suggestion is to plot Figure 7 for individual system (i.e., don't mix the output from the control and reduced loads experiments).

18. Figure 6. Statistic tests are needed to test if the differences in nutrient retention are significant between the control and reduced load experiments. I strongly suggest the authors use daily output instead of monthly mean to increase the sample size.

19. Line 165. Could you please provide the definition of "retention capacity," as it sounds like professional jargon to me?

20. Line 167. "…while the phosphorus retention capacity remains largely independent of load variations". This conclusion is drawn from Figure 7d. However, according to Figure 6b, P relative retention capacity seems to change significantly (need statistic test) when nutrient loads are halved. That is, Figure 6b contradicts Figure 7d. Please also see the comment 17.

21. Section 3.4. What is the purpose of this section? Is that designed to find the minimum resolution that can simulate the retention capacity well enough? If so, you already have the 150-m model and there is no need to try coarser ones.

22. Line 172. Running a 5550-m model may not be meaningful, as the resolution of the parent model is 2000 m. Instead, the authors can add a finer resolution test (maybe at 50 m).

23. Section 4.1 looks like another result section but lacks in-depth discussion. It reduplicates what has been shown in section 3.

24. Lines 185-186. There is no evidence shown to support this statement. Please see the comment 8.

25. Line 193-197. More quantitative analysis is needed to address the contribution of various sink terms to the net fluxes. Such analysis may ask for modification of model parametrization. As shown by DIN validation, the model also failed to capture the peak value in high-DIN period, which may result from the overestimated nutrient uptake rate by phytoplankton.

26. Line 200-202. Please see the comment 12. The authors may need to compare the contribution of sediment oxygen consumption and water stratification to the bottom DO changes.

27. Line 202. Please pinpoint the Grobes Haff in the map.

28. Lines 204-205. This study is not a hypoxia study. If there is a great discrepancy found between modeled and observed DO, then I suggest the author focus more on the nutrient retention.

29. Lines 206-207. This is a very strong statement. I've seen a low-trophic model with 11 phytoplankton functional groups.

29. Lines 208-209. I am not sure if it is true for the Baltic Sea. As I learnt, parameter tuning is usually needed for most ecosystem model when study region is changed due to the changes in multiple ecosystem aspect, like dominate species, lower-trophic complexity, and pollution conditions. So, I would suggest the authors be cautious when saying "without parameter tuning".

30. Lines 212-216. Isn't it obvious?

31. Lines 240-241. This may not be true. The authors should test the significance of P changes due to reduced nutrient loads. Also see the comments above.

32. Lines 244-245. To my understanding, it is not correct.

33. Line 245. Usually, at water surface, oxygen decreases as primary production decreases.

34. Figure 10. Are the changes between the control and reduced loads experiments? Please clarify it in the caption.

35. Figures 9 and 10. Need similar plots for P. Please also make the line styles and line colors consistent for the same term in both plots. Please also use the same name for the same terms.

36. Figure 11. Please clarify how the water residence time is calculated in the main text. It is important to show it because there are at least two definitions of water residence time as I know.

37. Lines 268-270. This conclusion confuses me. Do the authors mean that the riverine nutrient loads are mainly control by riverine water discharges rather by riverine nutrient concentration?

38. Figure 12. In (a) N and P loads decrease in recent years. Such negative trends may be contributed by (1) negative trend in riverine water discharges and (2) nutrient reduction actions in recent years. It is very interesting to compare these contributions to see if the human's efforts in nutrient reduction matter regarding water quality improvement.

39. Section 4.4. I am not sure why the authors are interested in the interannual signal. As shown by Figure 12a, the range of discharges and nutrient loads increase in recent years (e.g., ranges summarized every 5 moving years). It is likely due to the climate changes which cause more extreme events like droughts and floods. So, it would be more interesting to discuss the climate-change induced uncertainty in nutrient loads and the related implication.

40. The conclusion needs to be updated according to the revised contents.

41. Figure A1. Legends and text in the figure are hard to read. The color for the normalized RMSD is hard to see as well. And I believe RMSE (root-mean-square error) should be a more appropriate term to use when comparing model output and observations.

---

## Community Comment (CC1)

The manuscript "Transformative processes in the Oder Lagoon as seen from a model perspective" presents and validates a new model setup for the Oder Lagoon in Northern Germany. Further it assesses the nutrient retention in the Lagoon and thus adds to our understanding of the supply of nutrients from the Oder River to the Baltic Sea. The role of coastal zones as nutrient filters is an important and often overlooked topic, and especially the large nutrient flow from the Oder is important for the eutrophication of the Baltic Sea.

I suggest accept after minor revisions

Major comments:

I suggest removing the part on resolution (section 3.4). While interesting, I find that the role of resolution does not quiet fit with the general scope of the paper, and that the topic deserves further elaboration in a different study.

I suggest adding a figure/table on the seasonal variability of the different nutrient fractions (e.g. DIN, DON, PON) delivered to the Baltic Sea. From Figure 4, it looks like both DIN and DIP are very low in summer, suggesting a large seasonal variability in the fractions delivered to the Baltic. From a management point of view, the bioavailability of the nutrient supply is very important, especially if the bioavailability of the nutrients is very low during summer.

Discussion: Some thoughts about the relevance of the study for management of nutrients would be interesting. If targets are already reached, how were these targets defined, and did enough knowledge even exist regarding supply of nutrients from the Oder to the Baltic Sea? Are the targets too low if they were already reached?

Add section about nutrient cycling in the lagoon in the conclusion. The complete lack of this seems odd as it is the main focus of the manuscript.

Minor comments:

Title: I would suggest adding the word "nutrients"

L9: "Relative" nitrogen decreases

L25: What does "pollution" refer to

L47: Skogen et al., 2024 and 2021 have discussions on models versus observations.

L55: Also, it adds error to biogeochemical models of the Baltic Sea

L58: high"ly" resolved

L60: I would remove this

Figure 1a: A km-scale would improve understanding. Also the a thicker line at the coastlines would make the plot easier to see

L98: What was the spin-up time for water and sediment, how were the variables initialized. Do you have any knowledge on realistic sediment concentrations?

Figure 2 and 4: If the plots were made wider, the variability would be easier to see

L114: Is this due to lack of observations during peak times? Some discussion on the number of observations per year would be nice. It is very difficult to see from the plots how well the seasonal cycle is covered.

Figure 5: It took me a while to understand the difference between the figures. The title of b could be changed to something like "Mean duration of anoxic periods"

L198: What are the consequences of underestimation of DIP in summer, for productivity and the nitrogen cycle, including nutrient retention.

L249ff: To me the "," and "." In the numbering is confusing. European convention would be 1000 = 1.000,0 right? Not sure if the journal wants it differently, otherwise I would suggest not using comma to mark thousand.

---

## Author Comment (AC1)

We are very grateful for this comment from the community and will consider the numerous suggestions a revised version of our manuscript.

In the following, we respond to the remarks. Remarks are shown in blue and our response in black italic.

The manuscript "Transformative processes in the Oder Lagoon as seen from a model perspective" presents and validates a new model setup for the Oder Lagoon in Northern Germany. Further it assesses the nutrient retention in the Lagoon and thus adds to our understanding of the supply of nutrients from the Oder River to the Baltic Sea. The role of coastal zones as nutrient filters is an important and often overlooked topic, and especially the large nutrient flow from the Oder is important for the eutrophication of the Baltic Sea.

I suggest accept after minor revisions

Major comments:

I suggest removing the part on resolution (section 3.4). While interesting, I find that the role of resolution does not quiet fit with the general scope of the paper, and that the topic deserves further elaboration in a different study.

*Absolutely, also in line with the RCs, we decided to remove 3.4 and postpone it for an upcoming study.*

I suggest adding a figure/table on the seasonal variability of the different nutrient fractions (e.g. DIN, DON, PON) delivered to the Baltic Sea. From Figure 4, it looks like both DIN and DIP are very low in summer, suggesting a large seasonal variability in the fractions delivered to the Baltic. From a management point of view, the bioavailability of the nutrient supply is very important, especially if the bioavailability of the nutrients is very low during summer.

*We are going to present the riverine forcing in more detail in a revised version (see our reply to the RCs).We thank for the suggestion to present DIP, DIN and POM separately. Indeed, the bioavailability of POM and DOM is an uncertainty in the model forcing. Since no reliable information on bioavailability for the Oder River is available, we do not account for this in this study. However, this point is worse to discuss.*

Discussion: Some thoughts about the relevance of the study for management of nutrients would be interesting. If targets are already reached, how were these targets defined, and did enough knowledge even exist regarding supply of nutrients from the Oder to the Baltic Sea? Are the targets too low if they were already reached?

*The final decision about targets is a political one. The process of defining targets for rivers, coastal waters, and the open sea occurred separately. Therefore, reaching targets for a river system is not necessarily sufficient to achieve the targets for coastal waters. We believe that these targets need to be harmonized.*

Add section about nutrient cycling in the lagoon in the conclusion. The complete lack of this seems odd as it is the main focus of the manuscript.

*We will revise the section structure.*

Minor comments:

Title: I would suggest adding the word "nutrients"

*We will elaborate this suggestion.*

L9: "Relative" nitrogen decreases

L25: What does "pollution" refer to

*We need to clarify that we focus on nutrients.*

L47: Skogen et al., 2024 and 2021 have discussions on models versus observations.

*We will consider these additional references.*

L55: Also, it adds error to biogeochemical models of the Baltic Sea

L58: high"ly" resolved

L60: I would remove this

*Yes, see our previous comment.*

Figure 1a: A km-scale would improve understanding. Also the a thicker line at the coastlines would make the plot easier to see

*Will be done for Fig. 1a*

L98: What was the spin-up time for water and sediment, how were the variables initialized. Do you have any knowledge on realistic sediment concentrations?

*We started with an initialization from a Baltic Sea model with 1n.m. horizontal resolution. The spin-up for the Oder Lagoon model was 10 years. Sediment concentrations with a spatial coverage are to our knowledge not available.*

Figure 2 and 4: If the plots were made wider, the variability would be easier to see

*Regarding to the RCs, we will extend the validation which includes revised figures.*

L114: Is this due to lack of observations during peak times? Some discussion on the number of observations per year would be nice. It is very difficult to see from the plots how well the seasonal cycle is covered.

*Usually, one observation per month is available. We will show it in the revised manuscript.*

Figure 5: It took me a while to understand the difference between the figures. The title of b could be changed to something like "Mean duration of anoxic periods"

*Will be done for more clarity.*

L198: What are the consequences of underestimation of DIP in summer, for productivity and the nitrogen cycle, including nutrient retention.

*We will investigate it in detail in an upcoming study. In recent years, these peaks became quite rare. It appears that these peaks are a relic from former eutrophication.*

L249ff: To me the "," and "." In the numbering is confusing. European convention would be 1000 = 1.000,0 right? Not sure if the journal wants it differently, otherwise I would suggest not using comma to mark thousand.

*We will check it with the journal's requirements and remove commas if applicable.*

---

## Author Response (AR1)

**Review #1**

First of all, we would like to thank the referees and a comment from the community for the thorough reviews of our manuscript. We provide our response on how to consider the referee's suggestions in a revised version of our manuscript.

In the following, we respond to the referee's #1 remarks. Remarks are shown in blue and our response in black italic. Changes made in the manuscript are in red.

In general, the manuscript is well written, well presented and clear, including mathematical expressions and equations. It reaches innovative and robust conclusions that can greatly improve model studies in the area. Especially as little is known about coastal retention in the Baltic Sea. It is also relevant for policy measures, because official nutrient reductions do not account for nutrient retention in coastal waters. However, the conclusions could be more specific and, with additional discussions, could be more relevant for the entire Baltic Sea. While their methodology is valid, clear and relevant, it is difficult to be applied by any other institute studying the Baltic Sea's biogeochemistry, as setting a coastal model for one single region is costly (manpower, budget, time) and not all may have that possibility. However, by expanding their analysis further and adding more discussions on how the Order Lagoon compare to other coastal areas in the Baltic Sea, approximations for N and P retention could be estimated for other areas as well. Published results for the Baltic Sea mentioned in the paper have large discrepancies with specific coastal studies, and therefore new suggestions will be relevant to address including approximations for other coastal areas, especially for those similar to the Oder Lagoon. Their conclusions do mention that the method can be adapted for other regions beyond the Baltic Sea, but as written now is too vague. If this means adapting their model to a new area, it could translate as a costly task (time and computational) and not necessarily straightforward. In this sens, a better clarification on what is possible to extrapolate to other areas with this method is required. In other words, the main missing discussion in this manuscript is: Can the relation found in this study between loads and nutrient retention be used to estimate %retention in other basins? What can be used from these results to improve retention estimates in other coastal areas in the Baltic Sea and other Seas?

The Scientific significance of this study is good and after some revisions, will contribute significantly to our understanding of nutrient retention in coastal waters. The latter is poorly known and therefore, the content of this manuscript is very valuable. The methodology is clear and results relevant to biogeochemistry.

The scientific quality of the study is fair, the methods and approach are valid and innovative. However, it misses additional discussions, described in the specific comments below.

The presentation quality is good, the structure and figures are clear and well explained. However, it misses some additional figures, for example, in the validation section time series of model and observed oxygen should be shown, as well as more on the water column dynamics in the model. The oxygen section can also be expanded, as it is a major parameter affecting both N and P.

Section 4.4 should be expanded or moved to the results section. For details, see specific comment below.

In short, I would strongly recommend this manuscript for publication after minor/medium revision.

*We greatly appreciate the positive consideration of our manuscript and the comprehensive suggestions for its improvement. In the revised version, we will expand the validation section. We plan to extend Appendix A and present a summary of the findings in the main text.*

*Additionally, the referee has provided valuable advice for enhancing the discussion and conclusion sections. Our detailed responses to these comments can be found in the "specific comments" section.*

We have extended the validation section and included additional data from both observations and model simulations. The comprehensive validation section has been moved to the appendix. In the main text, we provide a summary of the validation and refer readers to the appendix for further details.

We have added a paragraph in the discussion (Section 4.2) to share our perspective on the extent to which our findings could be applicable to other regions.

Additionally, we have expanded the discussion in Section 4.4 to offer a broader perspective on upcoming changes in the Oder Lagoon.

Specific comments

Introduction

line 21: What is meant by extended residence time? Long? If so, replace by "long"

*"long" seems to be the right term.*

Changed into "long".

line 26: Please expand and add references. Why are there only a few coastal waters that can reduce nutrient loads in the Baltic Sea? Why not all coastal areas have that capacity? The Baltic Sea has many different types of coastal waters, please mention them as well and why would they be less effective in nutrient retention.

*Asmala et al., 2017, which we refer to, give a good classification of different Baltic Sea coastal types and potential filter function. We will summarize the Asmala et al. findings in this context.*

We introduced a paragraph on different coastal types and their retention capacity.

Line 28: Why is the Order Lagoon the most critical?

*"likely the most critical" We based this statement on Asmala et al. 2017, who identified the Oder Lagoon as the one with the highest denitrification rates. We will make it more clearly in a revised version.*

More references have been included supporting this statement. The most pronounced function is removal of DIN due to denitrification depending on residence time.

Line 48: Are the MAI here referred per basin in the Baltic Sea or to the entire Baltic Sea?

*We refer to MAIs in general, since they neglect a possible coastal filter. MAIs are based on loads into the "central" basin.*

It has been clarified in the text.

Line 55: Models can account for nutrient retention in the Baltic Sea taking values from literature and making assumptions for unstudied coastal areas. They can give good enough nutrient concentration near coastal areas, but can certainly be improved. The problem is not that they do not adequately account for it, but that there is not enough information to adequately account for all different types of coastal areas of the Baltic Sea. In this regard, this study can really improve such model results. Here, I would suggest to focus more on the fact that there is missing information on this for the Baltic Sea (and other Seas) and discuss this further. Importantly, this has been already mentioned in previous modeling studies, for example in:

Eilola et al., 2011, they mention that "the major differences between the nutrient supplies to the different models are due to the different assumptions on the bioavailability of phosphorus loads", referring to the retention of P in coastal waters. They also have a dedicated chapter on bioavailable nutrient loads in their discussions, which should be mentioned in this study.

Ruvalcaba-Baroni et al., 2024, they refer to the little information available and what can be currently used: "As the response to nutrient removal of different coastal types is poorly quantified... these factors are taken from previous studies in the Baltic Sea (Eilola et al., 2011; Edman and Anderson, 2014; Asmala et al., 2017)." In their discussion, they also mention that "one factor affecting detritus is the fraction of the organic matter coming from rivers that is actually bioavailable and not directly retained in coastal waters." They assume a constant fraction for the entire Baltic Sea and say that "the input of organic matter from rivers, especially nitrogen, could be improved by better accounting for river-specific organic matter retention in coastal waters".

Adding this discussion, will make the point of this study stronger, as it "partly" answers a problem that has been existing since many years.

*We would like to emphasize the importance of distinguishing between the non-bioavailability of riverine organic matter (OM) and the bioavailable OM and nutrients that are retained or filtered in coastal zones. This distinction is crucial at the process level, as these components respond differently to changing boundary conditions. Indeed, the understanding of nutrient load transformation and retention between monitored rivers and export from the coastal zone remains limited.*

*Altogether, the referee raised an important point that we will address in the revised manuscript.*

We added a paragraph in the introduction which point to the problem of "purely quantified" nutrient retention in coastal waters as the referee suggested.

Results

line 112: The authors mention that "the model fails to reproduce the low salinities observed between 2004 and 2008, possibly due to an underestimation of runoff in the forcing", but do not show the forcing used. What evidence do you have for this? It could also be due to the atmospheric forcing (precipitation vs evaporation). Please add more explanation for this, a supporting figure with, for example the runoff used in the high resolution model and how it compares to observations could be helpful here (or in supplementary). Also, why not show salinity and temperatures profiles or some water column performance?

*In the extended validation section (see response to general remarks), we will also consider vertical profiles. The suggestion that runoff is the cause is more of an educated guess. The meteorological forcing data comes from a consistent reanalysis, while the runoff data is based on observations from the Oder River. The model reproduces similar low salinities, for example, in 1999 and 2010. Therefore, we concluded that the runoff during the winter seasons of 2004-2008 is underestimated. We will attempt to find more robust arguments to explain the elevated salinity. Additionally, we will present the river data used in our analysis.*

We have added an extended validation section and discussed the reasons for model biases more thoroughly. Runoff was previously illustrated in the former Fig. 12a. However, we have introduced an additional runoff figure. The Oder River accounts for 98% of the river discharge. Atmospheric precipitation minus evaporation (P-E) is on the order of 1% of the discharge. Both freshwater sources (precipitation and evaporation) have negligible impact on salinity.

Observations are only available for the surface and near-bottom layers. Therefore, we are unable to present observational profiles. The mixed layer depth derived from model simulations is presented and provides insights into stratification properties.

Line 114: The authors mention that "However, this discrepancy has yet to be confirmed". This statement falls a bit short. How would you confirm this and why not doing it in this manuscript? Please rephrase or add information.

*We believe that we cannot confirm this in the manuscript. Observations are available at a monthly resolution at best and may miss extreme values. We will rephrase this statement accordingly.*

The new validation section (appendix) discusses model biases and possible reasons in more detail.

Line 123: The authors say "the model successfully captures the decreasing trend in winter phosphate". Plotting the trend directly in the figure for both model and observations would make this point more clear. It will also be relevant to plot it for the other biogeochemical parameters. The significance of the trend can then be estimated, which, for example, it is not entirely clear for chlorophyll-a at KHM (is the decreasing trend significant or not?). This information would be valuable for the discussions later on and for line 129, where the model and observed chla are described. From Fig 4, it looks like the model has also a more pronounced decreasing trend than the observations at KHM, but very good match at C.

*In the extended validation section, we will include trend lines and statistical estimates. The referee is correct; providing this information will strengthen the discussion.*

We added a trend line in the time series figure if the trend is significant.

Line 131-132: The authors mention that "phytoplankton response to changing ambient light conditions and (???) may vary between 23 and 60". I think the authors missed writing "Chl:C ratios". Please add the missing part. If so, would the Chl:C ratio be that different between C and KHM? Would this alone explain the discrepancies in model performance between C and KHM?

*We will re-phrase the sentence.*

We corrected the sentence referring to Chl:C ratios. Performance discrepancies between stations C and KHM, we cannot explain beyond some speculations.

Oxygen dynamics (section 3.2)

This section is very relevant to the manuscript. While the model shows oxygen depletion, there is missing evidence on how depleted the oxygen is in the area. Why not add oxygen time series in bottom waters from both observations and model? Is there any oxygen measurements in this area?

*Observations for oxygen are available. However, near-bottom observations are affected by the measurement process itself. Factors include how close the oxygen sensor can be lowered to the floor and whether the water column is disturbed by the measuring platform (vessel). Experimental evidence supporting these limitations is provided by Fredriksson et al. (2024). We will present both simulated and observed bottom oxygen concentrations, acknowledging that the observations may not accurately represent the true near-bottom oxygen levels.*

Available oxygen data are presented in the new validation section. However, observations near the bottom suffer from the measurement technique which we discuss as well.

This requires further discussions either to show more evidence for oxygen depletion or mention the lack of observations and what are the alternatives to "validate" the model in such case. What are the references for oxygen depletion in this area?

This is also relevant for the statement that the authors write in line 126 "..., the amount of phosphate released is insufficient to significantly increase the surface concentration." This is linked to oxygen, but also to mixing. Neither are shown. Water column dynamics can be shown in different ways (e.g., salinity and temperature profiles, mixing water depths, etc). It is difficult to judge how well the model represents the oxygen concentration in bottom waters or phosphorous retention in the sediments. It may as well be that the release of phosphorus is not insufficient in the model, but that the lower values in the surface are due to a biased runoff forcing that stratifies the water column too much in the model. Please add water column dynamic information (it can be in the appendix) and evidence for low oxygen concentrations in bottom waters from observations (e.g., time series for C and KHM of bottom oxygen).

We present oxygen from bottom water observations and discuss the inherent difficulties using these data.

In figure 5, an additional map showing the actual model oxygen concentrations in bottom waters for the months with oxygen depletion and available observations will be valuable.

We show bottom oxygen in the appendix for both available stations.

Line 164: Phosphorus retention in the sediments is also affected by bottom oxygen concentrations. It will be interesting to plot retention rates vs oxygen concentrations in bottom waters, especially for P. Could this (partly) explain the lack of relationship between rates of retention and P loads?

A high retention rate could impact the phosphate release during anoxia. However, too strong retention on the other hand would yield too low phosphate concentrations during oxic conditions.

Unfortunately, we do not have the retention rate directly in our diagnostic model output. Instead, we show the phosphate liberation together with near bottom oxygen.

Lines 199-205: Relates to comment above on Oxygen Dynamics. This paragraph does not have any reference. How do we know the Order Lagoon experiences anoxia? Oxygen is a relevant topic for this type of study. Please expand the discussions on this and evidences for oxygen depletion available for the Order lagoon.

*Aim of the model application was to show that the model sufficiently well represents the seasonal dynamics and the long-term development in the Oder lagoon. However, during our study, we observed short-term ecological effects where the model was not well in agreement with the monitoring data. One aspect is hypoxia and the other, related aspect, are the sudden summer peaks of inorganic phosphorus. Mass balances clearly indicate that the P-peaks have to result from a release from the sediment, very likely under anoxic conditions (Fe bound P), the so-called, internal eutrophication. The monthly monitoring data taken one meter above the ground does not indicate hypoxia. On the other hand, observed mussel and fish-kills, beside internal eutrophication, are clear indications for hypoxia and even anoxia. We decided to tackle this complex short-term problem in a separate paper, that is presently under preparation, because of all the implications for management and ecosystem assessment. Scientific literature indicating hypoxia and internal eutrophication in the Oder Lagoon does not exist.*

*Here we would expand the discussion and refer to the observed mussel and fish-kills as well as add literature from comparable lagoon systems that underpin the likelihood of hypoxia and internal eutrophication.*

*We will show water column dynamics and discuss possible consequences.*

We added a figure showing model stratification and phosphorus release from the sediment.

Line 175: How can the higher resolution model overestimate nitrogen retention?

*In the coarse-resolution model, the near-bottom oxygen concentration is overestimated. This enhances the coupled nitrification-denitrification process in the sediment, consequently increasing nitrogen retention.*

Impact of model resolution is postponed to another study.

Discussion

Line 185: The discussion relates to runoff forcing and would benefit from a runoff figure (as mentioned in comments above for line 112).

*As responded to line 112 comment, we will show river data.*

Runoff has been shown already in former Fig. 12a. However, we introduced another runoff figure.

Line 195: See comment above on Oxygen dynamics (section 3.2). This discussion should be expanded.

*Additional results from the extended oxygen analysis will be discussed.*

Oxygen concentration and its impact on phosphate release is discussed in the discussion section and the validation section.

198: There is much more literature available for P recycling in the Baltic Sea that can help tuning the model. Please add more references.

*We thank the referee for this advice.*

We indicate that the 2D sediment module's memory is too short to remember the eutrophication period over a longer time and to respond with high phosphate liberation to anoxia. Instead, it stores the history of a few years. Thus, it is not a matter of parameterization but of the model structure.

Line 205: So there are observations? Why not show them?

*Some are available but with the restrictions discussed in Fredriksson et al. 2024. We will show them.*

We show them now.

Line 220: Pleas add "As done in Eiola et al., 2011 and Ruvalcaba-Baroni et al., 2024, where they use a factor to account for nutrient retention" or similar (see comment above for line 55).

*We can do so, but we should be aware that bioavailability and retention/filtration are distinct processes.*

Introduction has been extended with the known uncertainties in availability of riverine loads.

Line 221-222: Please expand. This has more potential and general suggestions on how to improve model estimates in the Baltic Sea can be given based on the relevant results in this manuscript. The big picture would be to say something on how these very nice results (based on chapter 4.2, 4.3 and 4.4) can be used by the larger scientific community and perhaps even policy. This could be discussed in a chapter on its own (4.5?).

*We will consider including an additional section.*

We introduced a new paragraph in section 4.2 summarizing relevant fidings for retention and transferability to other regions.

Line 222: As mentioned before, not all coastal areas can be modeled the way it was done in this manuscript (it will be nice, but it will take time). Please expand on how your results could differ in other areas, based on bathymetry, residence time, oxygen content, etc and how they relate to published values (some of which are already mentioned in the introduction, but there are a few more, see discussions in Eilola et al., 2014). Or is it that every single coastal area in the Baltic Sea needs to be modeled separately to be able to estimate their coastal nutrient retention? Perhaps, but some relationships must be similar.

*We believe that the most crucial coastal regions can be simulated with reasonable effort. Asmala et al. (2017) provide a useful classification for determining which regions are most important. We will address this issue in the discussion.*

We discussed our opinion on what could be transferred to other regions and which coastal types require additional studies.

Line 240: There is no statistics shown in the paper, how do you know it is "not statistically significant"?

*We did this analysis and will show it in a revised manuscript.*

We show the statistics now.

Line 258: This is an important statement that could be repeated in the conclusion.

*Will be done.*

This part of the conclusions now.

Line 291: Please replace "." by ","

*Will be done.*

Done.

Interannual variability of discharge and loads and its consequences (4.4)

This entire section is structured more as results than as discussions. Please expand (add discussion and references) or move to results.

*The referee is correct. We will extent the discussion on this topic.*

Section 4.4 is extended by an additional paragraph.

Line 308: Please replace "emissions" by "inputs"

*Will be done.*

Done.

Line 310: How do you know that interannual variability is likely controlled by extreme events? Is there a reference or some evidence? It may seem obvious, but the relative or quantitative impact remains quite unknown for the Baltic Sea, as far as I know. Also what do you mean by "extreme events"? Storms, heatwaves and/or what else?

*We will clarify the type of event we are considering. A detailed analysis is beyond the scope of this study and is the subject of our ongoing research.*

We addressed the kind of extremes and possible impact in the additional discussion on this topic.

Line 315: How easy/difficult would it be to expand it to other regions in and outside the Baltic Sea? Please clarify.

*As we have already argued, we believe that additional studies can be conducted with reasonable effort. A prerequisite is the availability of reliable runoff data, which are accessible for most regions of the Baltic Sea.*

An additional paragraph on our opinion of transferability in section 4.2 is added.

320: I am not sure that the word "realistically impossible" can be used in this study, as it does not assess economical impacts and costs. Maybe rephrase to something less strong.

*We will consider this advice.*

We rephrased it to "unrealistic".

Figures

Figure 1. It would be good to mention already in the caption the navigation channel. It is quite obvious that there is a channel, but is not redundant to mention it. It would be also interesting to know when was this channel made (before the 60s?) to have an idea on how far back in time we should go to get "pristine conditions" in this area and to better interpret the observational data sets in this area as it seems to have a large impact for nutrient retention.

*We will modify the figure, give additional information of the channel and provide references. Modifications to the Swina River date back to 1721 and in 1880, a shortened and deepened artificial channel was completed. Subsequent, deepening projects included an increase to 9.6 meters in 1939 and to 10.5 meters in 1984 across the entire lagoon. Between 2018 and 2023, the entire waterway across the Oder Lagoon was deepened to 12.5 meters (Schernewski et al. 2025, https://doi.org/10.3390/environments12020035).*

We modified the figure and text accordingly.

Figure 5. Please add the period mean in the caption.

*We will add this information. The mean annual of anoxic days is based on the years between 1995 and 2019.*

We added this information.

*Ref.:*

*Fredriksson et al., 2024:* https://aslopubs.onlinelibrary.wiley.com/doi/10.1002/lno.12607

*Schernewski, G.; Neumann, T.; Piehl, S.; Swer, N.M. Ecosystem-Model-Based Valuation of Ecosystem Services in a Baltic Lagoon: Long-Term Human Technical Interventions and Short-Term Variability. Environments 2025, 12, 35. https://doi.org/10.3390/environments12020035*

**Review #2**

First of all, we would like to thank the referees and a comment from the community for the thorough reviews of our manuscript. We provide our response on how to consider the referee's suggestions in a revised version of our manuscript.

In the following, we respond to the referee's #2 remarks. Remarks are shown in blue and our response in black italic. Changes made in the manuscript are in red.

**General comments:**
In the manuscript by Neumann et al., the authors discussed the function of the Oder Lagoon in nutrient retention using a 150-m 3D ecosystem model. The lagoon was found to significantly affect the Baltic Sea eutrophication given the high nutrient retention capacity by the estuary (12% of riverine phosphorus and 40% of riverine nitrogen). However, the lagoon itself was found in a highly eutrophic condition which requires further attention and studies. The manuscript is well presented, clear, and easy to follow. However, as a model paper, the model evaluation section is too weak, lacking assessment of some key feature like, oxygen concentration, chl-a concentration (against satellite), and T/S vertical profiles. In addition, there is a lack of in-depth discussion in nutrient retention. I will list my comments in detail as followed.

Therefore, I recommend a major revision before publication.

*We greatly appreciate the consideration of our manuscript and the comprehensive suggestions for its improvement. In the revised version, we will expand the validation section. We plan to extend Appendix A and present a summary of the findings in the main text.*

*Additionally, the referee has provided valuable advice for enhancing the discussion and conclusion sections. Our detailed responses to these comments can be found in the "specific comments" section.*

We have extended the validation section and included additional data from both observations and model simulations. The comprehensive validation section has been moved to the appendix. In the main text, we provide a summary of the validation and refer readers to the appendix for further details.

We have added a paragraph in the discussion (Section 4.2) to share our perspective on the extent to which our findings could be applicable to other regions.

Additionally, we have expanded the discussion in Section 4.4 to offer a broader perspective on upcoming changes in the Oder Lagoon.

**Detailed comments:**
1. Line 27. Please add references for the examples.

*Efficiency of the coastal filter: Nitrogen and phosphorus removal in the Baltic Sea.*
*https://doi.org/https://doi.org/10.1002/lno.10644,*
*Biogeochemical Budgets of Nutrients and Metabolism in the Curonian Lagoon (South East Baltic Sea): Spatial and Temporal Variations. https://www.mdpi.com/2073-4441/14/2/164*

*Modeling the long-term dynamics of nutrients and phytoplankton in the Gulf of Riga*
*https://www.sciencedirect.com/science/article/pii/S0924796311000704*
*Modelling nutrient retention in the coastal zone of an eutrophic sea*
*https://bg.copernicus.org/articles/13/5753/2016/bg-13-5753-2016.pdf*
*Nutrient Retention in the Swedish Coastal Zone https://www.frontiersin.org/journals/marine-science/articles/10.3389/fmars.2018.00415/full*
*Biogeochemical functioning of the Baltic Sea: https://esd.copernicus.org/articles/13/633/2022/esd-13-633-2022.html*

We added several references.

2. Line 49. Could you be more specific about what the definition of "a good environmental status" are?

*We will add a reference for definition.*
*MSFD:*
*https://environment.ec.europa.eu/topics/marine-environment/descriptors-under-marine-strategy-framework-directive_en*
*In comparison to the good ecological status after WFD:*
*https://www.eea.europa.eu/en/analysis/indicators/ecological-status-of-surface-waters*
*Schernewski et al. (2015),*
*https://www.sciencedirect.com/science/article/pii/S0308597X14002358?via%3Dihub*

We added references for the definition of the "good environmental status".

3. Line 61. Why did the authors study the inter-annual variability instead of other scales like seasonality which should be more apparent than inter-annual signal for many earth systems?

*The seasonal signal is certainly more pronounced. However, for nutrient export to the Baltic Sea and its impact on the environmental status of the Baltic Sea, total loads over longer time scales are important. Here the focus is on the long-term variability. In a recent paper we studied the effects of shorter-term variability:*
*Ecosystem-Model-Based Valuation of Ecosystem Services in a Baltic Lagoon: Long-Term Human Technical Interventions and Short-Term Variability https://www.mdpi.com/2076-3298/12/2/35*

*In this study, it became obvious that the present temporal resolution of the input data, especially the available monthly load data of the Oder River limits the accuracy of the model hindcast. This is especially true for the Oder Lagoon, since it is strongly controlled by external Oder River loads. This analysis requires a modified model approach, a detailed field data analysis and a spatial analysis within the lagoon. This goes beyond the possibilities of this paper. Therefore, a separate paper is in preparation that studies the spatio-temporal seasonality in the lagoon and especially the role of extreme events such as droughts and floods as well as hot seasons.*

An analysis of short-term and seasonal variability will be addressed in a future publication. The focus of this study is on the lagoon's nutrient filtration function, which operates on a longer time scale.

4. Section 3.1. As a model paper, I think it is not enough to evaluate surface T, S, nutrient, and chl-a concentration over such big area, especially as the authors pointed out that water stratification was found as a factor to bottom dissolved oxygen deficiency. Evaluations of T/S vertical profiles, nutrient

profiles, oxygen profiles, and chl-a spatial patterns (against satellite estimates) are crucial to validating the modeled results.

*We will extend section A (appendix) with a detailed validation including vertical profiles and additional state variables.*

*Satellite products with a sufficient spatial and temporal resolution are only available since 2024 (https://data.marine.copernicus.eu/product/OCEANCOLOUR_BAL_BGC_L3_NRT_009_131/description), a time period not covered by the model. Further, satellite products are struggling with coastal waters like the Oder Lagoon, due to the higher turbidity.*

*The Oder Lagoon shows a strong patchiness of phytoplankton with respect to vertical location and small-scale horizontal distribution of phytoplankton. Satellite data can provide only a limited insight, has very limited absolute reliability and will be taken into account as soon as we study spatial effects in the lagoon, in detail.*

The validation section has been significantly expanded and moved to the appendix. Satellite imagery products, particularly for chlorophyll, are not available in reasonable quality for the inner coastal waters of the Baltic.

5. Figure 2. Does the model represent sea ice dynamics? I found that the modeled water temperature in winter is > 0 oC. As the authors mentioned at Line 113, the SST observations were not available in winter due to sea ice coverage. So, in reality, there should be sea ice covered the lagoon in winter. Please clarify the model's capability in sea ice modeling.

*Indeed, we use a coupled sea-ice-ocean model. We will mention this in the model setup section.*
The sea ice model is noted in the model description.

6. Figure 2 caption. "Modelsimulation" should be "Model simulation".

*We will correct this.*

Done.

7. Lines 110-114. More quantitative comparison is needed like providing R2, RMSE, and relative RMSE.

*The revised validation section will include this analysis.*
The validation section provides these quantities.

8. Line 111. As I observed, salinity was overestimated at Station C, but was underestimated at Station KHM between 2004 and 2008. Also, the authors need to provide evidence to support "possibly due to an underestimation of runoff in the forcing". The runoff forcings are from observations. If the statement is true (i.e., runoff is underestimated), it may indicate that the number of river point sources are not enough, or the atmospheric precipitation is underestimated. Please show the evidence to support the causes of underestimation of salinity.

*We will attempt to evaluate the reasons for the discrepancies. However, conducting extensive sensitivity studies is not feasible. We consider these discrepancies to be relatively minor and they do not affect the*

*vertical column properties, which we will present in the revised manuscript. If we cannot confirm that runoff uncertainties are the main reason, we will remove this statement.*
*Since salinity is controlled by the water exchange with the Baltic Sea via the deep (10.5 m) Swina Channel, the results in the lagoon depend on external forcing (Baltic Sea model) as well as the assumed connectivity between both systems. Single inflow-events play a major role for salinity in the lagoon. Location C is in the channel and it can be assumed that regular chip traffic plays a role in mixing saline water along the channel. However, we consider the agreement between data and simulation of salinity as sufficient for this study, since the differences have only negligible effects on nutrient cycling and pelagic ecology.*

By accident, we loaded incorrect observations for Station C in the evaluation program and have revised the figure accordingly. We would like to thank the referee for their critical review. It now appears that the model underestimates salinity on some occasions at both stations.

9. Lines 118-120. Why there is such a great difference in transport through the Dziwna between your estimates and others?

*In reality, the Dziwna channel is a long and shallow channel until it reaches the Baltic Sea. We did not include the appendix of this channel in our model. We will note this in the revised version.*

We have addressed this issue in the text and recommend incorporating a realistic long channel in a revised model setup.

10. Figure 3. Is the plot based on monthly average? Please provide the way how the transport contribution is calculated.

*It is calculated as usually, velocity\*rho\*area. This calculation is done online, that is, for each model time step. The monthly average is performed afterwards.*

11. Lines 125-127. There is a lack of evidence to address (1) that low bottom DO triggers the release of iron-bound phosphate from sediment and (2) that the amount of phosphate released is insufficient to increase the surface concentration.

*We will demonstrate it in more detail in the validation section of the revised manuscript.*
*During our study, we observed that the model was with respect to some parameters not well in agreement with the monitoring data, especially when it comes to short-term ecological effects. One aspect is hypoxia and the other, related aspect, are the sudden summer peaks of inorganic phosphorus. Mass balances clearly indicate that the P-peaks have to result from a release from the sediment, very likely under anoxic conditions (Fe bound P), the so-called internal eutrophication. This process is strongly influenced by the pollution history of the lagoon, namely the amount of Fe-P stored in the sediment at the beginning of the simulations 1995 (5-10 years after the pollution peak). This requires a different approach. We decided to tackle these complex aspects of anoxia and internal eutrophication in a separate paper and here focus on long-term developments.*

We show the relation between bottom oxygen concentration and liberation of phosphate from the sediment and hypothesize that our two-dimensional sediment module cannot store enough phosphorus for a later release.

12. Line 139. I am not convinced that the deeper channel acts as a sediment trap. Instead, I believe water stratification is the primary factor contributing to bottom DO depletion in the main channel. First, surface chl-a concentrations do not show a distinct pattern inside versus outside the main channel (Figure A1e). This suggests that the amount of sinking organic matter should follow a similar spatial distribution, especially in such a shallow lagoon (< 10 m), where organic matter does not drift far from where it sinks.

*The navigation channel acts as a sediment trap. Sedimentary material is continuously (by animals) or as an event (storm) re-suspended, transported horizontally by currents, and eventually ends up in deeper areas such as the navigation channel. (See figure below.)*

[Figure]

*Figure 1: Carbon content in model sediment Sep. 2005*

*There is little literature assessing the effect of dredging in the channel:*
*Ecosystem Services Supporting Environmental Impact Assessments (EIAs): Assessments of Navigation Waterways Deepening Based on Data, Experts, and a 3D Ecosystem Model https://www.mdpi.com/2073-445X/13/10/1653*

*Unfortunately in German only, a diploma thesis compiles data and makes a budget of dredging which clearly demonstrates the "sediment trap" property of the channel.*
*Minning, M. Der Schifffahrtskanal im Oderhaff. Eine Sediment-, Nähr- und Schadstofffalle?.Diplomarbeit. Christian-Albrechts-Universität, Kiel, 2003. Available online: https://eucc-d-inline.databases.eucc-d.de/files/documents/00000695_Diplomarbeit_Minning2.pdf.*

We added a reference referring to continuous dredging in the channel.

Second, water column stratification plays a crucial role in the development of bottom hypoxia, as demonstrated by numerous hypoxia studies. As the authors later mention, the model does not account for mixing processes due to heavy traffic in the main channel, which leads to discrepancies between the modeled and observed bottom DO concentrations (hypoxia is modeled but not observed in the measurements). This suggests that bottom hypoxia in the main channel may be more influenced by strong (or overestimated) stratification.

*For anoxia both requirements are needed: (1) Oxygen consumption, in shallow water mainly by sediments, and (2) prevention of oxygen supply for example due to stratification. This is the case in our model. In the real system, heavy ship traffic to and from Sczezin harbor exists. These ships have a draught*

*close to the floor of the navigation channel which regularly mixes the water column. This process is missing in the model.*

*For frequency of ship trafic see:*
*Ecosystem-Model-Based Valuation of Ecosystem Services in a Baltic Lagoon: Long-Term Human Technical Interventions and Short-Term Variability* *https://www.mdpi.com/2076-3298/12/2/35*
*About 3300 cargo ship arrive in Szczecin harbor per annum, which are about 18 cargo ships per day.*

13. Figure 4. Statistics like R2 and RMSE should be provided.

*See our response to 7.*
Comprehensive validation statistics are provided.

14. Lines 151-157. I think this is the core of this work. The authors may need to provide a diagram for quantification of each source and sink terms for both N and P. I found the Figures 9-10 attempt to address it, but there is a lack of quantification for the P sources and sink terms. Also, it is better to move Figure 9-10 to here.

*We will think about a revision of figures 9-10. Currently, we think about an additional table.*
We revised the figure and added a table of sources and sinks for N and P.

15. Line 157. As I understand, denitrification occurs at anoxic conditions. That is, denitrification rate at sediment should decrease as oxygen concentration increases.

*Denitrification occurs around the redoxcline. In the case of oxic bottom water, the redoxcline is located in the sediment. The denitrification at the sediment redoxcline is much more efficient as in the water column. The small spatial distance supports the coupled nitrification-denitrification.*
Figure 6 demonstrates the contribution of pelagic and sedimentary denitrification.

16. Line 160. Why don't you use the daily mean to increase the sample size? As I observed, the sample size in Figure 7 is small which may weaken the conclusion drawn.

*Unfortunately, river load data are on a monthly basis. These data are needed for the relations with loads and the relative retention.*

17. Figure 7. The plot read confused to me. The authors mix the output of the control and reduced load experiments when generating this plot. However, the plot includes two types of signals: (1) annual signal of the retention rate which changes as nutrient loads; (2) retention rate changes due to the changing system when nutrient is reduced. The former is the one what the authors want to analysis. However, regarding the latter one, when the total nutrient loads are reduced, the entire ecosystem will adapt to such changes and turn out to be a new system. For example, some phytoplankton species can become the dominate species given their higher adaptation to low-nutrient environment. Such changes may affect the sinking organic matter not just in spatial distribution but also in temporal phases. So, my suggestion is to plot Figure 7 for individual system (i.e., don't mix the output from the control and reduced loads experiments).

*We will use different colors for the experiment and analyze whether the results will change. However, we will explain the figure and consequences in more detail as the referee suggested.*

Former Fig. 7 (Fig. 5) is updated and show color coded the two different model experiments.

18. Figure 6. Statistic tests are needed to test if the differences in nutrient retention are significant between the control and reduced load experiments. I strongly suggest the authors use daily output instead of monthly mean to increase the sample size.

*See our comment to 16. We will improve the statistics.*
A statistical evaluation is added.

19. Line 165. Could you please provide the definition of "retention capacity," as it sounds like professional jargon to me?

*Retention capacity (in this context) refers to the ability to retain nutrients within a system. This term is also used in other contexts, such as the retention of water in soils or energy in a battery. We use it synonymously with filter capacity or function.*

20. Line 167. "…while the phosphorus retention capacity remains largely independent of load variations". This conclusion is drawn from Figure 7d. However, according to Figure 6b, P relative retention capacity seems to change significantly (need statistic test) when nutrient loads are halved. That is, Figure 6b contradicts Figure 7d. Please also see the comment 17.

*See our comment to 18. We will improve statistical analysis.*
Statistical analysis shows that the relative P retention is independent on loads which is given in the revised manuscript.

21. Section 3.4. What is the purpose of this section? Is that designed to find the minimum resolution that can simulate the retention capacity well enough? If so, you already have the 150-m model and there is no need to try coarser ones.

*We will remove section 3.4.*
Section 3.4 is removed from the manuscript.

22. Line 172. Running a 5550-m model may not be meaningful, as the resolution of the parent model is 2000 m. Instead, the authors can add a finer resolution test (maybe at 50 m).

*See our comment to 21, and we will not setup a 50m model.*

23. Section 4.1 looks like another result section but lacks in-depth discussion. It reduplicates what has been shown in section 3.

*We will revise sections 3 and 4.1 to reduce repetitions.*
We revised section 4.1

24. Lines 185-186. There is no evidence shown to support this statement. Please see the comment 8.

*See our comment to 8,* and what we did in response to comment #8.

25. Line 193-197. More quantitative analysis is needed to address the contribution of various

sink terms to the net fluxes. Such analysis may ask for modification of model parametrization.
As shown by DIN validation, the model also failed to capture the peak value in high-DIN period, which may result from the overestimated nutrient uptake rate by phytoplankton.

*We show our model results with reasonable care. Further "sensitivity studies" are not productive and beyond the scope of this study.*
In addition to our comment above, we do not see that the model fails predicting DIN.

26. Line 200-202. Please see the comment 12. The authors may need to compare the contribution of sediment oxygen consumption and water stratification to the bottom DO changes.

*See our comment to 12.*
We present stratification data and phosphate liberation (a proxy for oxygen deficiency) at station KHM. Unfortunately, temperature data are unavailable for station C, which hinders stratification analysis at this location.

27. Line 202. Please pinpoint the Grobes Haff in the map.

*Will be done.*
Done.

28. Lines 204-205. This study is not a hypoxia study. If there is a great discrepancy found between modeled and observed DO, then I suggest the author focus more on the nutrient retention.

*DO is reproduces by the model fairly well which we will show. An Exception is the navigation channel for reasons we explained in our comment to 12.*
Bottom oxygen concentrations are presented in the validation section. We discuss uncertainties stemming from both measurement limitations and the potential impacts of ship traffic.

29*a*. Lines 206-207. This is a very strong statement. I've seen a low-trophic model with 11 phytoplankton functional groups.

*In our opinion, model complexity depends on the scientific question.*

29. Lines 208-209. I am not sure if it is true for the Baltic Sea. As I learnt, parameter tuning is usually needed for most ecosystem model when study region is changed due to the changes in multiple ecosystem aspect, like dominate species, lower-trophic complexity, and pollution conditions. So, I would suggest the authors be cautious when saying "without parameter tuning".

*We document what we did. We will replace "open sea" with "Baltic Sea" which we simulate with the same model. The referee has a point; an altered model structure may require a re-calibration. However, we did not change the model.*

30. Lines 212-216. Isn't it obvious?

*Certainly yes, but usually one have to make a compromise between quality and costs. Our intention was to give some guidance for larger scale models. However we will remove section 3.4.*
Section 3.4 is removed from the manuscript.

31. Lines 240-241. This may not be true. The authors should test the significance of P changes due to reduced nutrient loads. Also see the comments above.

*See our comment to 14.*
We proofed the statistics.

32. Lines 244-245. To my understanding, it is not correct.

*See our comment to 15.*

33. Line 245. Usually, at water surface, oxygen decreases as primary production decreases.

*The sediment-water interface is meant, line 244-245.*

34. Figure 10. Are the changes between the control and reduced loads experiments? Please clarify it in the caption.

*Yes, the caption could be more precise.*
Caption has been modified.

35. Figures 9 and 10. Need similar plots for P. Please also make the line styles and line colors consistent for the same term in both plots. Please also use the same name for the same terms.

*We thank the referee for this hint.*
The figures have been modified to maintain consistent style and color schemes. Given that phosphorus has only one sink, a separate figure similar to that for nitrogen was deemed redundant. Instead, a table has been introduced (Table 1).

36. Figure 11. Please clarify how the water residence time is calculated in the main text. It is important to show it because there are at least two definitions of water residence time as I know.

*The total volume of the lagoon was divided by the river discharge. This simple calculation follows the international lake approach going back to Vollenweider (1976) and many subsequent publications. It allows the estimation of critival loads.*
*Vollenweider, R. A. (1976). Advances in defining critical loading levels for phosphorus in lake eutrophication. Memorie dell'Istituto Italiano di Idrobiologia, 33, 53–83.*
We clarified how residence time was estimated.

37. Lines 268-270. This conclusion confuses me. Do the authors mean that the riverine nutrient loads are mainly control by riverine water discharges rather by riverine nutrient concentration?
Indeed, this represents our conclusion regarding the interannual and multiyear variability of nutrient loads. However, it is important to note that this conclusion may not apply to long-term perspectives, particularly when significant changes in the catchment begin to take effect.

38. Figure 12. In (a) N and P loads decrease in recent years. Such negative trends may be contributed by (1) negative trend in riverine water discharges and (2) nutrient reduction actions in recent years. It is very interesting to compare these contributions to see if the human's efforts in nutrient reduction matter regarding water quality improvement.

*Yes, thank you, we have to rephrase it, to better make clear that the concentration of both nutrients, N and P, is largely independent from river discharge. This has been observed already earlier and is the reason why the long-term assessment of critical loads used a discharge correction/normalization (e.g. Friedland et al. 2019). However today, we see the tendency of a climate change induced generally reduced annual water discharges. This means in recent years, your point (1) is more important than (2), which dominated the load reductions in the 1990's. We can elaborate a bit more on it, because a tendency to more extreme floods can partly counteract that.*

*Friedland, et al. 2019: [https://doi.org/10.3389/fmars.2018.00521,2019](https://doi.org/10.3389/fmars.2018.00521,2019).*
As clarified in our response to this comment above, we have addressed the raised concern regarding load changes.

39. Section 4.4. I am not sure why the authors are interested in the interannual signal. As shown by Figure 12a, the range of discharges and nutrient loads increase in recent years (e.g., ranges summarized every 5 moving years). It is likely due to the climate changes which cause more extreme events like droughts and floods. So, it would be more interesting to discuss the climate change induced uncertainty in nutrient loads and the related implication.

*See earlier comment: yes, therefore, a separate paper is in preparation that studies the spatio-temporal seasonality in the lagoon and especially the role of extreme events such as droughts and floods as well as hot seasons.*

*We show and discuss that high and low runoff adequately modifies nutrient loads. A relation to climate warming cannot be established from our relatively short simulation period.*
We have incorporated a paragraph discussing the potential impacts of climate change. However, a comprehensive analysis of the effects of extreme events and climate warming will be the subject of a future study.

40. The conclusion needs to be updated according to the revised contents.

*Yes.*
The conclusion is updated.

41. Figure A1. Legends and text in the figure are hard to read. The color for the normalized RMSD is hard to see as well. And I believe RMSE (root-mean-square error) should be a more appropriate term to use when comparing model output and observations.

*We agree that the colors in Figure A1 are hard to distinguish and will adjust the figure in the revised manuscript. Using the normalized RMSD instead of the RMSE gives us the possibility to make the numbers comparable between the station and parameters. Further, the normalized RMSD gives us a good measure for the model skill, as values below 1 indicate that the standard deviation of the observations is higher than the RMSE, meaning that the model results stay within the natural variability.*

We use a modified figure in the revised manuscript.

---

## Referee Report (RR1)

This manuscript used a high-resolution coastal model to study the retention capcity of the Oder Lagoon. The topic is interesting, however, the manuscript need a major revision before publication for the following major concerns.

First, as a model study, the manuscript lacks sufficient details regarding model description and needs clarification: 1) How are model parameters determined? Please provide justification or references for parameter values.  2) In oxygen-rich environment, phosphate becomes bound to iron oxide. How does the model simulate iron oxide? Are this process and phosphate release simulated prognostically or through parameterization? 3) In this model, the light attenuation is determined by chlorophyll and CDOM. Please explain how to simulate CDOM and how well is the simulation of CDOM? 4) For riverine input, only fresh water and nutrients were described. How about other state variables, including CDOM, each functional group of phytoplankton, and etc? Since the liminic phytoplankton thrive in fresh and turbid water, I assume that the fresh water is turbid with high chlorophyll or CDOM concentration. Then, the question is, how are the riverine inputs of chlorophyll (each groups of phytoplankton) and CDOM specified, by observations or some assumptions? 5) Does meterological forcing include data of nutrient deposition? How is the nutrient deposition simulated in the model? 6) The authors state in Line 104: "the extracellular excretion of dissolved organic matter by phytoplankton results in non-Redfield carbon uptake". This sentence is unclear to me and needs clarification. Also, I would like to ask how does the model deal with nutrient stoichiometry? Is it fixed or variable in the model? 7) Some model components are missing in the model description, including the ice model, the two-layer sediment model, and the definition of residence time. The authors answered in their response letter that they had added description of the ice model in the revised manuscript, however, it remained absent. For the residence time, please give the equation used to calculate the residence time in the manuscript.

Second, the model validation is not sufficient enough to support their conclusions. 1) The model failed to reproduce the observed season cycle of surface chlorophyll (one peak versus two peaks). Since phytoplankton growth is the primary driver of organic matter deposition and therefore is a key in N/P retention, the authors should discuss how this model bias might affect their key conclusions. Attributing the model bias to the fixed carbon-to-chlorophyll ratio is insufficient. 2) The model significantly underestimated bottom oxygen concentration. However, the authors' justifications are unsatisfactory:

(i) mismatch in depth between model and observations – the validation should use the closest model grid cells to measurement depths for comparisons

(ii) ship-induced mixing – The authors claim in appendix that the model is more realistic and ship-induced mixing as a shortcoming of observations. I disagree with

this statement. While ship disturbances occure in reality, neglecting this process is the a weakness of the model, not shortcoming of observations.

Since the oxygen controls phosphate release and denitrification, the model's failure in simulating bottom oxygen cast doubt on conclusions of this manuscript. Please improve the model performance or provide discussions on whether this bias affect conclusions. 3) There is no validation of bottom nutrients, which is an important indicator of phosphate release and nutrient retention. 4) For model validation of stratification, Figure A7 is hard to read. Please provide some metrics (e.g. RMSE, R-square) between observations and modelled results.

Third, the current version of discussion section should go to results, and some discussions looks not highly relevant. The subsection 4.2 should be shortened. In addition, some key issues are not discussed in the current mansucript. For instance, (1) as I mentioned above, How might model bias in simulating surface chlorophyll and bottom oxygen affect the nutrient retention conclusions? What sensitivity analysis can be done to test the robustness of comclusions? (2) Differences in N- and P-retention response to nutrient loads deserve explanation, including but not limited to model bias in simulating bottom oxygen, the N:P ratio of riverine input (limited nutrients), and the inherent differences between N and P cycles. Answers to this question may help clarify whether this conclusion in Odor Lagoon can be applied to other coastal systems. (3) What is the implications for larger-scale models? In the introduction part, the authors mentioned that some baltic model accounted for the filter function of nutrients by assuming bioavaliability or reduction factors. The authors should compare their finds to previous empirical approaches, and discuss the validity of the previous assumptions.

Finally, some comments and concerns raised in the initial review were not well addressed and some revisions promised in the response round were not made in the revision round. These will be listed in the detailed comments. To facilitate the evaluation of revisions, I would also suggest the authors to 1) povide locations for each revision in their response letter and 2) include revised text in the response letter.

Detailed comments:

L87: please provide the number of model grid cells or the length/width of the lagoon. This will help readers who are not familiar with this region to better understand the model resolution.

L104: Why this lead to non-Redfield ratio of nutrient uptake?

L128: How to quantify/calculate mas transport?

Section 3.1: wrong reference to Fig1

L217-228: The N-retention rate (40%) in this study is much higher than previous studies (<30%). Why? The authors attribute this difference to the fact that previous studies only account for denitrification while neglecting other source/sink terms; however, this explanation appears insufficient. Based on Figure 6, the non-denitrification processes contribute minimally to the overall nitrogen budget.

Response to reviewer #1

Comment on L199-205: the promised discussion of hypoxia were not made in the revised manuscript.

Comment on L222: Regarding "what could be transfered to other regions", this was not well addressed. The authors present several assertions yet without providing any justification or supporting evidence in the manuscript. Please also see my major comments above. In addition, the promised discussion on "which coastal type requires additional studies" were not made.

Response to reviewer #2

Comment 5: no description of ice model in revised manuscript

Comment 9: maybe I missed it, but I can't find it in the revised manuscript

Comment 10: please add this to manuscript

Comment 12: This didn't answer the reviewer's concerns.

Comment 18: No statistical evaluation in the manuscript. The reviewer was requesting a statistic test to examine differences between the control and halved-load experiments, rather than the trends from combined experiments.

Comment 19: see major comments above

Comment 33: The response is unclear

Comment 36: see major comments above

Comment 38: It confused me. The authors kept mentioning that changes in river discharge reflect changes in nutrient load because nutrient concentrations were stable, yet at the same time argued that this relationship did not apply for long-term analyses.

---

## Referee Report (RR2)

The manuscript has been improved now. However, there remains a few comments not well addressed.

Regard model description, while I understand that not all details need to be included, the paper should be self-explained. For critical model components, a breif explanation is need followed by reference to previous papers, rather than requiring readers to find answers in another paper without any hints. The authors should do a better job of explaining their model for the reader. For example:

- 1. Sediment Processes: The text mentions the binding and release of PO4 with iron oxides. However, it is unclear whether this is just a description of a natural process or if your model actually simulates it. There also lacks discussion of this mechanism in the following part of this manuscript. Also, it seems this process is simulated by a sediment model, but this sediment model is not mentioned until the Discussion section. This makes it very hard for most readers unless they already know your work well.
- 2. **Atmospheric Deposition:** Similarly, it is still not stated whether the model's forcing data includes atmospheric deposition fluxes of nitrogen and phosphorus. The authors should clearly state which external drivers are included in their model.

**Regarding model validation,**

- 1. The appendix still states that the model results are more realistic than the observations. Although this was acknowledged and corrected in the response letter, the corresponding text in the manuscript has not been revised.
- 2. The authors note that a key reason for the bottom dissolved oxygen (DO) underestimation is that observations are from the lowest ~1 meter, while the model output is from the lowest 20 cm. However, they keep using the 20 cm model data for comparison. This doesn't make sense as it produces severely underestimated oxygen levels (reaching hypoxic levels), which would significantly alter the simulated fate of N and P in the bottom water. It would be more scientifically sound to use model data comparable to the observed depth (1 m). Currently, the 1-meter comparison is only provided for one station as appendix; please add this comparison for the other station as well.
- 3. Concerning the validation of stratification, if the authors refuse to include a statistical analysis, please improve the figure. In their current form, it is difficult to assess whether the model correctly simulates the stratification process.

Regarding sections 3.3, it would be better if this section ended after you show the long-term changes in nutrient loads. The parts that come after would fit much better in the Discussion section. In addition, you can also connect these results with your earlier

finding in Section 3.2 about the relationship between load and retention, and stengthen the Discussion by addressing the following points:

- 1. The implications of the long-term load trends for nutrient retention within the system.
- 2. A comparison of your findings with larger-scale models that apply a fixed retention value: do your results suggest these models tend to overestimate or underestimate retention? This discussion should be clearly and explicitly
- 3. A direct discussion on the limitations of the fixed retention approach used in previous larger-scale models, in light of your evidence that retention is load-dependent. This is a key motivation for your study (as stated in introduction) but currently lacks sufficient discussion.

---

## Author Response (AR2)

**Review #3**

First of all, we would like to thank the referee for the thorough review of our manuscript. We provide our response on how to consider the referee's suggestions in a revised version of our manuscript.

In the following, we respond to the referee's #3 remarks. Remarks are shown in black and our response in red, and text modifications in blue.

Some of the reviewer's concerns stem from the perceived brevity of the model description. However, all components of our model system have been thoroughly introduced in previous publications, to which we have provided appropriate references. Rather than replicating extensive details from these sources, we have opted to provide concise explanations of each model component's purpose. Including full descriptions would unnecessarily expand the manuscript. That said, we will enhance certain aspects of the model description in the revised version to improve clarity where needed.

This manuscript used a high-resolution coastal model to study the retention capcity of the Oder Lagoon. The topic is interesting, however, the manuscript need a major revision before publication for the following major concerns.

First, as a model study, the manuscript lacks sufficient details regarding model description and needs clarification: 1) How are model parameters determined? Please provide justification or references for parameter values.

In line 81 of our manuscript, we refer to Neumann et al. (2022) for a detailed explanation and validation of the model. The ERGOM model employs over 130 parameters, which—consistent with standard practice in ecosystem modeling—were determined during the calibration phase to optimize model performance. Since the complete model description is provided in the cited reference, we consider it unnecessary to repeat these details here. Furthermore, we contend that, within a reasonable range, an ecosystem model's performance depends more critically on its structural design than on the specific parameter values selected.

2) In oxygen-rich environment, phosphate becomes bound to iron oxide. How does the model simulate iron oxide? Are this process and phosphate release simulated prognostically or through parameterization?

Iron oxide complexes are represented as a prognostic state variable in the model. As with all ecosystem models, not every process is described deterministically; instead, certain aspects are parameterized. For a detailed treatment of these processes, we refer the interested reader to our comprehensive model description in Neumann et al. (2022).

3) In this model, the light attenuation is determined by chlorophyll and CDOM. Please explain how to simulate CDOM and how well is the simulation of CDOM?

In line 94 of our manuscript, we refer to Neumann et al. (2021) for a detailed description and validation of the optical model, including colored dissolved organic matter (CDOM). As CDOM observations for the Oder Lagoon are unavailable, we used Secchi depth measurements as an alternative validation approach.

4) For riverine input, only fresh water and nutrients were described. How about other state variables, including CDOM, each functional group of phytoplankton, and etc? Since the liminic phytoplankton thrive in fresh and turbid water, I assume that the fresh water is turbid with high chlorophyll or CDOM concentration. Then, the question is, how are the riverine inputs of chlorophyll (each groups of phytoplankton) and CDOM specified, by observations or some assumptions?

As described by Neumann et al. (2021), CDOM is introduced into the model via riverine runoff based on observational data. All phytoplankton groups in the model are capable of growth when environmental conditions are favorable. Consequently, there is no need to include marine phytoplankton groups in the runoff. For limnic phytoplankton, however, we employ a different approach: a portion of the total nutrient input is allocated specifically to limnic phytoplankton. This distinction should be clarified in a revised version of the manuscript.

**Additional text:**

A minor fraction of the total nutrient loads enters the lagoon through the limnic phytoplankton state variable, which ensures seed concentrations near the river mouth. Riverine CDOM concentrations are prescribed using a monthly climatology, as described in detail by Neumann2021.

5) Does meterological forcing include data of nutrient deposition? How is the nutrient deposition simulated in the model?

Atmospheric deposition is realized as a boundary condition (air-sea fluxes) based on data provided by HELCOM which are originated from EMEP (https://www.eea.europa.eu/data-and-maps/data/external/emep-n-atmospheric-deposition). We will add this information in a revised version of the manuscript.

**Additional text:**

Atmospheric deposition is realized as a boundary condition (air-sea fluxes) based on data provided by HELCOM assessments (e.g., HELCOM, 2018) which are originated from EMEP (https://www.eea.europa.eu/data-and-maps/data/external/emep-n-atmospheric-deposition).

6) The authors state in Line 104: "the extracellular excretion of dissolved organic matter by phytoplankton results in non-Redfield carbon uptake". This sentence is unclear to me and needs clarification. Also, I would like to ask how does the model deal with nutrient stoichiometry? Is it fixed or variable in the model?

The non-Redfieldian carbon uptake and resulting stoichiometric relationships are described in Neumann et al. (2022). In the revised manuscript, we will include essential information to ensure readers understand the fundamental mechanisms of this process.

**Text changed:**

Removed: Furthermore, the extracellular excretion of dissolved organic matter by phytoplankton results in non-Redfield carbon uptake.

Added: Furthermore, phytoplankton excrete extracellular dissolved organic matter with non-Redfield stoichiometry, resulting in non-Redfield carbon uptake, while maintaining canonical Redfield ratios within their cellular composition.

7) Some model components are missing in the model description, including the ice model, the two-layer sediment model, and the definition of residence time. The authors answered in their response letter that they had added description of the ice model in the revised manuscript, however, it remained absent. For the residence time, please give the equation used to calculate the residence time in the manuscript.

In response to Referee #2's request regarding the model's sea ice capabilities, we confirm that our manuscript specifies both the sea-ice model employed and its foundational reference. Our implementation utilizes a regional configuration of the well-established MOM (Modular Ocean Model) framework, which includes a fully coupled ocean-sea-ice component.

As this sea ice component is based on a widely-used, thoroughly documented framework with numerous existing publications detailing its formulations and capabilities, we have chosen to reference the foundational publications rather than replicate these descriptions. The key reference providing the basic formulations of the sea-ice model is Winton (2000), which offers comprehensive details on its physical and thermodynamic representations.

We believe this approach provides sufficient information for readers to understand our implementation while avoiding unnecessary duplication of well-documented methods. However, we are happy to add a brief summary of the sea ice model's key features in the revised manuscript if the referee finds this would be helpful for clarity.

Text added:

The sea ice component implements:

- a) A three-layer vertical thermodynamic scheme
- b) Multiple ice thickness categories with dynamic redistribution
- c) Category transition mechanisms responding to thermodynamic and mechanical forcing
- d) Full ice dynamics incorporating internal stresses via an elastic-viscous-plastic rheology

The equation and relevant references for residence time estimation are provided in the caption of Figure 8 (Fig. 6 of the revised manuscript).

Second, the model validation is not sufficient enough to support their conclusions.

1) The model failed to reproduce the observed season cycle of surface chlorophyll (one peak versus two peaks). Since phytoplankton growth is the primary driver of organic matter deposition and therefore is a key in N/P retention, the authors should discuss how this model bias might affect their key conclusions. Attributing the model bias to the fixed carbon-to-chlorophyll ratio is insufficient.

We disagree with the characterization that our model fails in representing chlorophyll concentration. Organic matter production is also reflected in the nutrient dynamics, which likely provides a more accurate representation of net production than chlorophyll concentration. Furthermore, we maintain that the chlorophyll-to-carbon ratio has significant implications on the chlorophyll concentration estimate, contrary to the suggestion of minimal impact. We also acknowledge that water column opacity represents an additional influencing factor, which we will incorporate into the discussion of our revised manuscript.

**Text added:**

Another limitation of our simulations is the systematic underestimation of winter opacity (Fig. A4), which may potentially advance the timing of vernal blooms. This discrepancy likely arises from our model's current implementation, which accounts for resuspension of organic matter only, while neglecting the resuspension of mineral sediments. The omission of mineral sediment dynamics probably contributes to the simulated overestimation of winter water clarity.

- 2) The model significantly underestimated bottom oxygen concentration. However, the authors' justifications are unsatisfactory: (i) mismatch in depth between model and observations the validation should use the closest model grid cells to measurement depths for comparisons (ii) ship-induced mixing The authors claim in appendix that the model is more realistic and ship-induced mixing as a shortcoming of observations. I disagree with this statement. While ship disturbances occure in reality, neglecting this process is the a weakness of the model, not shortcoming of observations. Since the oxygen controls phosphate release and denitrification, the model's failure in simulating bottom oxygen cast doubt on conclusions of this manuscript. Please improve the model performance or provide discussions on whether this bias affect conclusions.
- (i) Unfortunately, the available observations from the Oder Lagoon are limited to surface and near-bottom measurements only, with no exact depth indication. With regard to the observing staff, near-bottom observations are estimated to represent conditions approximately 1 meter above the seafloor.

While we could include an additional figure comparing observed near-bottom oxygen concentrations (1 m above bottom) with model data at the corresponding depth, we believe the current figure showing simulated bottom oxygen remains the most appropriate representation for our analysis.

Near bottom oxygen observations and model simulation 1 meter above bottom at station KHM.

The figure presented above compares observed near-bottom oxygen concentrations with model simulations at approximately equivalent depths corresponding to the measurement locations. We emphasize that within the 10th to 90th percentile range, anoxic conditions never occur, thereby maintaining stable redox conditions. While anoxia represents a rare and transient phenomenon in our system, these episodic events nonetheless exert significant influence on phosphorus cycling and benthic community dynamics. This conclusion remains valid even when considering the closest near-bottom simulation values (20 cm above the sediment-water interface).

**Text added:**

Figure A3 illustrates the elevated oxygen concentrations simulated at 1 m above the seafloor (approximating the measurement depth) compared to our simulated near-bottom oxygen concentrations (Fig. A1f). We emphasize that within the 10th to 90th percentile range, anoxic conditions never occur, thereby maintaining stable redox conditions. While anoxia represents a rare and transient phenomenon in our system, these episodic events nonetheless exert significant influence on phosphorus cycling and benthic community dynamics. This conclusion remains valid even when considering the closest near-bottom simulation values (20 cm above the sediment-water interface).

(ii) We acknowledge the referee's valid point regarding unresolved ship traffic as a limitation of our model. To clarify, we never intended to suggest that the model provides enhanced realism in not representing ship traffic effects—this appears to be a misunderstanding that we will correct by revising the relevant text for greater precision. Furthermore, we explicitly identify this limitation as a model weakness and highlight it as an area for future improvement in the conclusions section of our manuscript.

The observed discrepancies between measured and simulated near-bottom oxygen concentrations are primarily confined to the navigation channel, where they result from the model's inability to account for ship traffic effects. However, given that the navigation channel constitutes only a small fraction of the

Oder Lagoon's total area, these localized oxygen dynamics have minimal impact on the overall phosphate release and binding processes in the lagoon system.

**Modified Text in the discussion section:**

Oxygen deficiency in the near-bottom water is widespread in the Oder Lagoon. The most affected area is the artificial navigation channel (see section 3.2). In the recent model implementation, the model does not account for ship traffic, which causes regular vertical mixing down to the bottom. Thus, in contrast to the model, observations do rarely show hypoxia in the navigation channel. Given that the navigation channel constitutes only a small fraction of the Oder Lagoon's total area, these localized oxygen dynamics have minimal impact on the overall phosphate release and binding processes in the lagoon system.

3) There is no validation of bottom nutrients, which is an important indicator of phosphate release and nutrient retention.

An additional figure will be provided in a revised version of the manuscript.

**Text added:**

For completeness, we have included Figure A5, which compares simulated and observed near-bottom nutrient concentrations at stations KHM. Notably, these near-bottom values exhibit minimal divergence from surface concentrations (Fig A1), suggesting frequent vertical homogenization of the water column. This interpretation is further supported by Figure A9, which demonstrates that stratification events in the Oder Lagoon are typically short-lived and frequently disrupted by meteorological forcing.

4) For model validation of stratification, Figure A7 is hard to read. Please provide some metrics (e.g. RMSE, R-square) between observations and modelled results.

We question the added scientific value of including this data, given that observational records are predominantly unavailable during the winter season, which would limit meaningful comparison and interpretation.

The primary objective of this figure is to demonstrate that stratification patterns are consistently observed in both field measurements and model simulations. Given that stratification constitutes a necessary precondition for anoxia development, this correspondence demonstrates that both the natural system and our model have the potential to develop anoxic conditions under appropriate circumstances. We will strengthen interpretations in the revised manuscript version.

**Text added:**

The comparative analysis presented in this figure reveals consistent stratification patterns between empirical observations and model simulations. Given that stratification constitutes a necessary precondition for anoxia development, this correspondence demonstrates that both the natural system and our model have the potential to develop anoxic conditions under appropriate circumstances.

Third, the current version of discussion section should go to results, and some discussions looks not highly relevant. The subsection 4.2 should be shortened.

We find this comment insufficiently specific regarding which particular elements require relocation. Nevertheless, we will carefully evaluate the organization of our Results and Discussion sections in the revised manuscript. Additionally, we will review Section 4.2 to eliminate any potential redundancy and improve overall clarity.

In response to referee's comments, we have restructured the manuscript by relocating the majority of interpretative content from the Discussion to the Results section. The former Section 4.2 has been condensed and streamlined for improved clarity and conciseness. All specific textual modifications are indicated in the marked-up version of the revised manuscript.

In addition, some key issues are not discussed in the current mansucript. For instance,

1) as I mentioned above, How might model bias in simulating surface chlorophyll and bottom oxygen affect the nutrient retention conclusions? What sensitivity analysis can be done to test the robustness of conclusions?

As outlined in our previous response, we do not consider the observed differences in chlorophyll time series to significantly impact net primary production estimates. This interpretation is supported by the consistent nutrient dynamics across observations and our simulations, which remain the primary driver of production.

In the conclusion section of our revised manuscript, we will propose potential avenues for future sensitivity analyses.

**Paragraph added:**

To complement our hindcast simulations, targeted scenario analyses could provide valuable insights into the system's sensitivity to specific anthropogenic interventions. Particularly informative scenarios might

- a) Assessments of varying ship traffic intensities and their impacts
- b) Evaluations of potential navigation channel deepening effects

Such scenario simulations would enable a more comprehensive understanding of the lagoon's response to management measures and environmental modifications.

2) Differences in N- and P-retention response to nutrient loads deserve explanation, including but not limited to model bias in simulating bottom oxygen, the N:P ratio of riverine input (limited nutrients), and the inherent differences between N and P cycles. Answers to this question may help clarify whether this conclusion in Odor Lagoon can be applied to other coastal systems.

We would like to clarify that the differences in retention capacity due to load reduction arising from nitrogen and phosphorus pathways are explained in Sections 3.3 and 4.2 of our manuscript. Based on our current understanding of the system, we maintain that no significant correlation between riverine N:P ratios and retention exists, as the Oder Lagoon is not typically nutrient-limited. This lack of limitation

suggests that retention processes are not primarily controlled by nutrient stoichiometry in this particular ecosystem.

(3) What is the implications for larger-scale models? In the introduction part, the authors mentioned that some baltic model accounted for the filter function of nutrients by assuming bioavaliability or reduction factors. The authors should compare their finds to previous empirical approaches, and discuss the validity of the previous assumptions.

It is important to note that most model descriptions in the literature do not explicitly quantify bioavailability parameters. These parameters are results of model calibration. By contrast, our reduction/retention factors are derived from a mechanistic approach. We believe this methodological distinction is evident in our manuscript.

Finally, some comments and concerns raised in the initial review were not well addressed and some revisions promised in the response round were not made in the revision round. These will be listed in the detailed comments. To facilitate the evaluation of revisions, I would also suggest the authors to 1) povide locations for each revision in their response letter and 2) include revised text in the response letter.

**Detailed comments:**

L87: please provide the number of model grid cells or the length/width of the lagoon. This will help readers who are not familiar with this region to better understand the model resolution.

The model's spatial resolution is specified in lines 87-88 of the manuscript. While the spatial extent can be approximated from Figure 1, we are happy to provide the exact domain dimensions in the revised version of the manuscript.

Text added:

(altogether 330x191 grid points)

L104: Why this lead to non-Redfield ratio of nutrient uptake?

We note that this comment appears to be identical to the referee's major concern #6. Therefore, we refer the referee to our response to that concern, where this issue has already been addressed.

L128: How to quantify/calculate mas transport?

Mass transport in our model is calculated using the standard oceanographic approach of integrating the product of density and velocity across the cross-section. As this is a fundamental fluid dynamics principle that is well-established in the literature, we have not included a detailed explanation in the manuscript.

Section 3.1: wrong reference to Fig1

The reference to Figure 1 is intended to direct readers to the locations of stations KHM and C. To prevent any potential misunderstanding, we will either: (1) remove the reference if it proves confusing, or (2) clarify it by adding a parenthetical note such as '(for station locations, see Figure 1)' in the revised manuscript.

**Text added:**

(for station locations, see Figure 1)

L217-228: The N-retention rate (40%) in this study is much higher than previous studies (<30%). Why? The authors attribute this difference to the fact that previous studies only account for denitrification while neglecting other source/sink terms; however, this explanation appears insufficient. Based on Figure 6, the non-denitrification processes contribute minimally to the overall nitrogen budget.

Given the known uncertainties in experimental denitrification estimates—compounded by their sparse spatial and temporal coverage—we consider the discrepancy between 40% and 30% to be within an acceptable range of variation. Our modeling approach does not aim to precisely replicate observations (which themselves contain inherent uncertainties), but rather to provide complementary insights through an alternative methodological framework.

We acknowledge the referee's valid point that sediment burial alone cannot fully account for the observed differences, and we will remove this statement in the revised manuscript.

**Text added:**

However, burial and pelagic denitrification are only minor contributions to the nitrogen retention.

Regarding nutrient retention, Pastuszak et al. (2005) reported values of 85% for total nitrogen (TN) and 72% for total phosphorus (TP). However, it is important to note that their study area included inland portions of the Oder Lagoon that fall only partially within our model domain. We will incorporate this reference in the revised version to provide additional context for our results.

**Text added:**

Pastuszak et al. (2005) report substantially higher retention rates of 85% for nitrogen and 72% for phosphorus. However, these elevated values must be interpreted with consideration of methodological differences: their study encompassed inland regions of the Oder Lagoon that extend beyond our defined model domain, potentially influencing the observed retention metrics.

Response to reviewer #1 Comment on L199-205: the promised discussion of hypoxia were not made in the revised manuscript.

To the best of our current knowledge, no direct observations of anoxic conditions in the Oder Lagoon have been documented. This absence of empirical data can primarily be attributed to suboptimal temporal and spatial sampling strategies. Nevertheless, several proxy indicators suggest episodic anoxia occurrence, including documented fish and mussel mortality events as well as summer phosphate peaks.

These findings have been comprehensively reported in Schernewski et al. (2025). We will synthesize and incorporate these proxy-based observations in the revised version of our manuscript.

Schernewski G, Neumann T, Piehl S and von Weber M (2025) New approaches to unveil the unknown: oxygen depletion and internal eutrophication in a Baltic lagoon over decades. Front. Environ. Sci. 13:1620191. doi: 10.3389/fenvs.2025.1620191

**Paragraph added:**

To the best of our current knowledge, no direct observations of anoxic conditions in the Oder Lagoon have been documented. This absence of empirical data can primarily be attributed to suboptimal temporal and spatial sampling strategies. Nevertheless, several proxy indicators suggest episodic anoxia occurrence, including documented fish and mussel mortality events as well as summer phosphate peaks. These findings have been comprehensively reported in Schernewski et al. (2025].

Comment on L222: Regarding "what could be transferred to other regions", this was not well addressed. The authors present several assertions yet without providing any justification or supporting evidence in the manuscript. Please also see my major comments above. In addition, the promissed discussion on "which coastal type requires additional studies" were not made.

In lines 253 et seq., we present our conclusions regarding the transferability of our findings to other regions, including an assessment of which regions may require further investigation.

Response to reviewer #2

Comment 5: no description of ice model in revised manuscript

We refer the referee to our detailed response to Major Comment 7 under Referee #3's review, where this specific issue has been thoroughly addressed.

Comment 9: maybe I missed it, but I can't find it in the revised manuscript

We direct the referee to lines 128, 180, and 186 of our manuscript, where this specific aspect is addressed.

Comment 10: please add this to manuscript

We have already addressed this question in our response to the review. Regarding inclusion in the manuscript, we respectfully maintain that this information constitutes established textbook knowledge and therefore does not require repetition in our publication.

Comment 12: This didn't answer the reviewer's concerns.

We are confident that our response to the review effectively addresses the referee's concern.

Comment 18: No statistical evaluation in the manuscript. The reviewer was requesting a statistic test to examine differences between the control and halved-load experiments, rather than the trends from combined experiments.

Upon reflection, we acknowledge that our initial response to this comment could have been more precise. We maintain that statistical proof of the retention difference between the two experiments is not scientifically justified in this context, as it was not the primary research question. Rather, our investigation focuses on determining whether different nutrient loads elicit distinct relative retention capacities—a question that can be effectively addressed through the trend analysis we have conducted and presented in our manuscript.

Comment 19: see major comments above

Regarding Comment 19, we have already provided a definition of retention in our response to Referee #2.

Comment 33: The response is unclear

We would like to clarify that Referee #2's comment pertains to the **water surface**, whereas our discussion specifically addresses the **sediment-water interface**, as clearly indicated in line 237 of the manuscript.

Comment 36: see major comments above

We kindly refer the referee to our response to Major Comment 7, where this issue is addressed.

Comment 38: It confused me. The authors kept mentioning that changes in river discharge reflect changes in nutrient load because nutrient concentrations were stable, yet at the same time argued that this relationship did not apply for long-term analyses.

We would like to clarify that our statement does not suggest this relationship fails to hold in long-term **analyses**. Rather, we emphasize that under a long-term **perspective**—particularly when considering potential changes in land-use or agricultural practices, as noted in the manuscript—additional factors may influence the observed dynamics.

---

## Author Response (AR3)

**Review #4**

First of all, we would like to thank the referee again for the thorough review of our manuscript and the helpful recommendations. We provide our response on how to consider the referee's suggestions in a revised version of our manuscript.

In the following, we respond to the referee's #3 remarks. Remarks are shown in black and our response in red, and text modifications in blue.

**Regard model description**, while I understand that not all details need to be included, the paper should be self-explained. For critical model components, a breif explanation is need followed by reference to previous papers, rather than requiring readers to find answers in another paper without any hints. The authors should do a better job of explaining their model for the reader. For example:

1. Sediment Processes: The text mentions the binding and release of PO4 with iron oxides. However, it is unclear whether this is just a description of a natural process or if your model actually simulates it. There also lacks discussion of this mechanism in the following part of this manuscript. Also, it seems this process is simulated by a sediment model, but this sediment model is not mentioned until the Discussion section. This makes it very hard for most readers unless they already know your work well.

Firstly, we wish to clarify that our sediment representation is properly characterized as a module rather than a fully resolved model. This component constitutes a highly parameterized, two-dimensional representation of early diagenetic processes. We explicitly acknowledge this simplified approach as a model limitation in the Discussion section, where we critically evaluate its implications for our simulation results.

**Text added to section 2 "Model Setup":**

The sediment module parameterizes key early diagenetic processes, including coupled nitrification-denitrification, organic matter remineralization, iron-phosphate complex formation/dissolution dynamics, and permanent burial. These processes are vertically integrated and represented through a two-dimensional model variable.

Under oxic conditions at the sediment-water interface, phosphate binds to ferric iron Fe3+ to form particulate complexes. Hydrodynamic erosion may subsequently resuspend these iron-bound phosphate particles, facilitating their transport via bottom currents to depositional zones. Conversely, under anoxic conditions, iron oxides undergo reductive dissolution, releasing phosphate into the overlying water column as dissolved inorganic phosphorus, following established redox-sensitive phosphorus cycling mechanisms (Neumann and Schernewski, 2008).

2. Atmospheric Deposition: Similarly, it is still not stated whether the model's forcing data includes atmospheric deposition fluxes of nitrogen and phosphorus. The authors should clearly state which external drivers are included in their model.

We appreciate the referee's suggestion and have attempted to address this comment by explicitly specifying "nitrogen and phosphorus" in line 131 of the revised manuscript.

**Changed text:**

Atmospheric deposition of nitrogen and phosphorus is realized as a boundary condition (airsea fluxes) based on data provided by HELCOM assessments (e.g. HELCOM) which are originated from EMEP (<a href="https://www.eea.europa.eu/data-and-maps/data/external/emep-n-atmospheric-deposition">https://www.eea.europa.eu/data-and-maps/data/external/emep-n-atmospheric-deposition</a>).

**Regarding model validation,**

1. The appendix still states that the model results are more realistic than the observations. Although this was acknowledged and corrected in the response letter, the corresponding text in the manuscript has not been revised.

While we maintain that our model's high-resolution near-bottom oxygen simulations (20 cm above seafloor) likely provide a more accurate representation of true benthic conditions than standard observations taken at 1 m depth above sea ground, we acknowledge the current limitations in empirical validation. The methodological challenges associated with traditional observational techniques at this depth have been previously discussed in our manuscript.

In the absence of direct, high-resolution observational data for near-bottom oxygen concentrations in this specific region, we have removed our previous definitive statements regarding oxygen dynamics. The only available evidence suggesting potential anoxic conditions in the Oder Lagoon remains indirect, as documented by Schernewski et al. (2025b), which we have appropriately cited.

**Removed text:**

We assume the model is closer to reality due to common shortcomings in observational techniques.

2. The authors note that a key reason for the bottom dissolved oxygen (DO) underestimation is that observations are from the lowest ~1 meter, while the model output is from the lowest 20 cm. However, they keep using the 20 cm model data for comparison. This doesn't make sense as it produces severely underestimated oxygen levels (reaching hypoxic levels), which would significantly alter the simulated

fate of N and P in the bottom water. It would be more scientifically sound to use model data comparable to the observed depth (1 m). Currently, the 1-meter comparison is only provided for one station as appendix; please add this comparison for the other station as well.

We recognize that this comment is directly related to the previous point regarding nearbottom oxygen representation. To comprehensively address the referee's concerns, we have made the following revisions:

- Included the model's 1-meter above bottom oxygen values in our validation analysis (directly comparable to observational data)
- Completely revised the corresponding paragraph

These modifications ensure our validation approach directly addresses the referee's specific concerns while maintaining scientific rigor in our oxygen dynamics assessment.

**Deleted text:**

The near-bottom oxygen levels exhibit clear differences between the model and observations. We assume the model is closer to reality due to common shortcomings in observational techniques. The model data are collected approximately 20 cm above the seafloor, while observations are typically taken from 1 m above the seafloor. Fredriksson (2024) demonstrate that strong oxygen gradients exist above the seafloor in coastal waters, which cannot be resolved with traditional CTD instruments. Furthermore, the measurement platform - typically a vessel - may disrupt the vertical structure of the water column.

**Added text:**

For our near-bottom oxygen comparisons, we utilized model data from 1 meter above the seafloor, corresponding directly to the standard observational measurement depth. However, we note that oxygen concentrations in our highest-resolution model output (20 cm above the seafloor) are systematically lower. This vertical gradient is consistent with recent findings by Fredriksson (2024), who demonstrated that strong oxygen gradients commonly exist in the bottom boundary layer of coastal systems - gradients that typically exceed the resolution capabilities of conventional CTD instrumentation. Additionally, we acknowledge that traditional measurement platforms (typically research vessels) may introduce artifacts by disrupting the natural vertical stratification of the water column during sampling operations.

Our model-data comparison at 1 m above the seafloor reveals a slight negative bias in simulated oxygen concentrations. Despite this, our analysis confirms that within the 10th-90th percentile range of observations, anoxic conditions are never encountered, indicating persistent oxic conditions and stable redox potential throughout most of the study period. While anoxic events remain rare and temporally limited in this system, these episodic occurrences exert disproportionate influence on:

- Phosphorus biogeochemical cycling
- Benthic community structure and function

Notably, these conclusions remain robust even when considering our highest-resolution near-bottom simulation data (20 cm above the sediment-water interface, not shown).

3. Concerning the validation of stratification, if the authors refuse to include a statistical analysis, please improve the figure. In their current form, it is difficult to assess whether the model correctly simulates the stratification process.

To enhance visual clarity and better illustrate seasonal stratification patterns, we have condensed the time series presentation. This focused representation more effectively highlights the pronounced seasonal variability in water column stratification, particularly the recurrent summer intensification that represents the period of highest potential for anoxic conditions.

**Added and deleted text:**

For visual clarity, we focus the analysis on the period from January 1997 to December 2000.

The comparative analysis presented in this figure reveals consistent stratification patterns between empirical observations and model simulations.

Our comparative analysis demonstrates strong agreement between observed and simulated stratification patterns, with both datasets showing peak stratification intensity during summer months.

**Regarding sections 3.3**, it would be better if this section ended after you show the long term changes in nutrient loads. The parts that come after would fit much better in the Discussion section. In addition, you can also connect these results with your earlier finding in Section 3.2 about the relationship between load and retention, and stengthen the Discussion by addressing the following points:

We appreciate the referee's guidance regarding the manuscript structure and offer the following clarifications regarding sections 3.2–3.4:

**Section References**

We suspect there may be a minor typographical clarification needed: the intended sections for revision are likely 3.3 and 3.4 (rather than 3.2 and 3.3), as these align more closely with the substantive content discussed.

**Structural Approach to Section 3.4**

While we acknowledge the suggestion to split Section 3.4 into separate Results and Discussion components, we have instead adopted an alternative approach that addresses the referee's earlier advice to relocate discussion-oriented content to the Results section. Specifically:

- We have retained the integrated presentation of findings in Section 3.4, as this maintains the logical flow of our analysis.
- To align with the referee's recommendation, we have moved the management-related implications (originally in Section 3.4) within the Discussion section, ensuring a clearer separation of results and broader interpretations.

This approach allows us to preserve the narrative coherence of our findings while addressing the referee's structural concerns.

As part of our restructuring of the Results and Discussion sections, we have incorporated a new subsection (4.3) titled 'Implications for water quality' within the Discussion:

**Implications for water quality**

Our 25-year simulation (1995-2019) indicates mean annual nutrient exports from the Oder Lagoon to the Baltic Sea of 28,988 tonnes for nitrogen and 1,945 tonnes for phosphorus. When accounting for these retention-mediated reductions, the resulting nutrient loads to the Baltic Sea consistently remained below the target thresholds established by the Baltic Sea Action Plan (BSAP) throughout the simulation period. However, the Oder Lagoon case demonstrates that achieving water quality targets in connected river systems and the Baltic Sea does not necessarily translate to good ecological status in transitional water bodies, highlighting the need for ecosystem-specific management approaches.

A good ecological status in the lagoon would require significant additional nutrient load reductions and the implementation of measures in the river catchment at high and likely unrealistic costs. It is likely that the lagoon must be regarded as a naturally eutrophic ecosystem with limited management possibilities. A re-evaluation of water quality targets in the lagoon requires a more detailed study that includes neighboring coastal waters to address interrelationships, relates model data to field data with a focus on the assessment stations, and carefully considers evaluation aspects such as water depth and evaluation period.

1. The implications of the long-term load trends for nutrient retention within the system.

We have relocated and rephrased the following paragraph from Section 4.1 to Section 4.2 of the Discussion, as its content is more thematically aligned with the latter section's focus:

"Coarse-grained models, typically employed for long-term simulations of the entire Baltic Sea basin, often fail to adequately resolve coastal hydrodynamic features and associated biogeochemical processing (i.e., the coastal filter function). For large river systems discharging into lagoons prior to reaching the open Baltic Sea, we therefore recommend utilizing load-corrected riverine inputs. Such corrections should ideally be informed by locally-specific retention estimates derived from high-resolution lagoon models. Where possible, these retention capacity assessments should further account for load-dependent variability in nutrient processing. efficiency."

**We added the following paragraph to 4.2:**

Current state-of-the-art approaches typically apply empirically-derived bioavailability factors, determined through model calibration, to account for coastal nutrient retention. However, these conventional methods present several limitations:

- They generally employ globally uniform factors that neglect the heterogeneity of coastal filter systems
- They assume temporal constancy, ignoring potential variability in retention efficiency

Our proposed mechanistic approach offers two key advantages:

- Enhanced regional realism: By explicitly quantifying lagoon-specific retention capacities using high-resolution local models, we generate more accurate, spatially-resolved nutrient load estimates
- Improved model performance: The mechanistic representation of retention processes facilitates more realistic ecosystem model calibration, potentially reducing compensatory errors in other model components

This mechanistic approach necessitates quantitative assessment of lagoon retention capacities through dedicated, high-resolution local modeling efforts. Particular emphasis should be placed on characterizing load-dependent variability in retention efficiency, as nutrient processing rates often exhibit nonlinear responses to input concentrations.

For comparable coastal filter systems, particularly lagoons, established empirical relationships (e.g., retention capacity as a function of water residence time) may serve as valuable initial approximations. Such relationships provide scientifically grounded starting points that can be subsequently refined through site-specific high-resolution local modeling effort.

2. A comparison of your findings with larger-scale models that apply a fixed retention value: do your results suggest these models tend to overestimate or underestimate retention? This discussion should be clearly and explicitly

We have addressed this comment through the additional paragraph included in our response to the previous point. Regarding the specific question of whether fixed retention values systematically lead to over- or underestimation:

The direction and magnitude of potential bias introduced by fixed retention values cannot be definitively determined a priori, as this depends on:

- The specific retention value selected (which may be higher or lower than actual sitespecific retention capacities)
- Spatial variability in coastal filter efficiency across different lagoonal systems
- Temporal dynamics in retention processes that are not captured by static values

This inherent context-dependency precludes a universal statement about bias directionality. Our mechanistic approach explicitly accounts for this variability, thereby reducing the uncertainty associated with fixed retention assumptions.

3. A direct discussion on the limitations of the fixed retention approach used in previous larger-scale models, in light of your evidence that retention is load dependent. This is a key motivation for your study (as stated in introduction) but currently lacks sufficient discussion.

We note that this concern has been comprehensively addressed in our response to the referee's first comment. The detailed explanation provided there regarding limitations of the conventional approach and advantages of our mechanistic modeling approach directly applies to this related point as well.

---

## Author Response (AR4)

**Review #5**

We sincerely appreciate the editor's thorough and constructive review of our manuscript. We have carefully considered each remark and recommendation. Implementing the revisions has significantly enhanced the quality and clarity of our work. Below, we provide point-by-point responses to the editor's comments, using the following color-coding for clarity:

- **Editor's remarks**: Black (original text)

- **Our responses**: Red

- **Text modifications**: Blue (tracked changes in the revised manuscript)

L3: For greater clarity, please specify the actual years this period represents.

We added: (1995 – 2019).

L10: It would be helpful to include a brief comment on whether phosphorus retention (the 12% average) was found to be stable or also varied with changing riverine loads.

We added: In contrast, phosphorus retention is independent on loads.

L15: Please specify under what conditions such achievement is unlikely. e.g., "...unlikely under the current management targets"?

We followed the editor's advice and added: under the current management targets

L68: The phrasing "improve the input" is ambiguous in this context. Do you mean "... improve the representation of organic matter inputs from rivers in the model"? Please clarify.

We followed the editor's advice and changed the sentence to: They also argue that river-specific organic matter retention factors in coastal waters would improve the representation of organic matter inputs from rivers in models.

L71: "a spatially high resolved…" -> "a high-spatial-resolution"

We followed the advice of the editor and changed the sentence accordingly.

L87: "mode" -> "model"

We corrected the typo.

L101: "light climate" -> "light field"?

We changed light climate into light field.

L143: The single-sentence paragraph is redundant and can be deleted. The structure of the section is already logically implied by the subsection titles.

We removed the paragraph.

L145: "reasonable" -> "reasonably"

We corrected reasonable to reasonably.

L145-155: Please cite the specific figures where the data (e.g., bottom oxygen, mass transport values, stratification) are presented.

We added references to figures of oxygen, mass transports, and stratification in the appendix.

L156: Please briefly specify the key variables assessed and provide a short summary of the overall conclusion from the model performance evaluation in the appendix. This will give readers context before they navigate to the supplementary material.

We restructured this subsection and added missing information. The text reads now:

Appendix A presents an expanded model performance analysis. We evaluate time series and climatology of hydrodynamic (temperature, salinity, stratification, and mass transport), and biogeochemical (nutrients, chlorophyll-a, and bottom oxygen) parameters.

The model reasonably reproduces the climatology as well as the interannual variability of nutrients, temperature, salinity at stations KHM and C (for station locations, see Fig. 1). For bottom oxygen (Figs. A1 and A2), the model predicts lower values than in the observations. Reasons are that (i) measurements are not as close to the bottom as the model data, (ii) the measurement platform (vessel) itself disturbs the stratification, and (iii) at station C commercial ship traffic induces strong vertical mixing which is not part of the model.

The mass transport through the Dziwna channel (for location, see Fig. 1) is elevated compared to known values (Fig. A6). The reason is the truncated Dziwna channel in the model reducing the hydraulic resistance and facilitating an enhanced discharge at the expense of a lower discharge through the Swina channel.

The stratification of the water column at station KHM (for location, see Fig. 1) compares well with observed stratification (Fig. A8). Stratification establishes as events, such as those in summer, while during winter the water is well mixed. Stratification results in oxygen deficiencies, which yield phosphate liberation from the sediment. This process is not directly observed but is indirectly indicated by elevated phosphorus concentrations in summer (Fig. A7).

Our evaluation confirms that the model reasonably represents the Oder Lagoon ecosystem's key dynamics, establishing it as a reliable tool for experimental analysis and system property investigation.

L160: Please provide the specific oxygen concentration thresholds used to define oxygen-depleted conditions and anoxia. e.g., < 2 mg/L or 1 mg/L?

Depleted conditions refer in our text to a zero oxygen concentration. We added (zero oxygen concentration) to the text.

L184-188: The statement that "a significant relationship between the retention rate and phosphorus loads does not exist" appears to be contradicted by Fig. 4b. The time series in Fig. 4b consistently shows higher relative P retention under the control run compared to the 50% load reduction, suggesting a load-dependent response. This conclusion is more accurately supported by Fig. 5d. The scatter plot in Fig. 5d shows no significant correlation when all data points are considered, which aligns with the stated lack of a significant relationship. Please clarify.

The editor is right. Both figures appear contradicting. We checked again the data base for these two figures and did not found any mistake. Indeed, we get a positive correlation between phosphorus load and relative retention as figure 4b suggests. However, this correlation is weak and the p-value of a Spearman rank test is 0.68 which confirms H0 that no correlation exists. The same test for nitrogen retention gives a p-value of zero and rejects H0. We noted this in the revised manuscript: A Spearman rank correlation test yields a non-significant regression coefficient (p = 0.68). This result fails to reject the null hypothesis of no correlation, thereby indicating that the apparent differences in the Fig. 4b are not statistically significant at conventional confidence levels.

L187-188: Please provide the result of this t-test (e.g., p-value).

See our comments to the previous editor's remark. In the revised version, we used a Spearman rank test to account for non-Gaussian data. However, the result is the same.

L190: Please provide statistical evidence to support the qualitative claim that the correlation for P "appears less pronounced" than for N in Fig. 6.

We changed the paragraph and provided the statistical indicators: Our analysis reveals a further relationship between nitrogen and phosphorus retention efficiency and the lagoon's water residence time (Fig 6). The relationship for phosphorus is less pronounced but statistically robust (p(N) = 0.0004, p(P) = 0.0017). In general, we can state that the longer water remains in the lagoon, the higher the relative retention of nutrients in the lagoon.

L193: Please clarify how the 40% and 12% retention estimates were derived. Are they the mean of the annual retention efficiencies calculated over the entire 25-year simulation period?

We changed the sentence for more clarity: The mean of the annual retention over the 25 years simulation period is approximately 40% for nitrogen and 12% for phosphorus.

L193-194: As commented earlier, the conclusion that "the phosphorus retention capacity remains largely independent of load variations" is inconsistent with Fig. 4b. The argument should be based on the lack of a significant correlative relationship in the scattered data of Fig. 5d. Please clarify.

We changed the sentence to: Nitrogen retention capacity increases with reduced lagoon nutrient loads, while phosphorus retention remains largely load-independent, statistically confirmed by analysis of Fig. 5d data.

See also our response to remark L184-188.

L246-247: Please cite the specific panel of the figure.

We specified the panel number. Fig. 7f.

Figures 5-7: For the regression analysis presented in Figs. 5-7, please report the associated p-values to assess whether the correlations are statistically significant.

We introduced a sentence how we evaluated the significance of correlations presented in the manuscript at the end of chapter 2. This should prevent replications throughout the text:

The observed statistical relationships were evaluated using Spearman rank correlation tests, with significance assessed via p-values. The null hypothesis (H0) posits no correlation between the examined variables. If the calculated p-value falls below the conventional significance threshold (alpha = 0.05), we reject the null hypothesis, thereby providing statistical evidence for a significant relationship between the variables.

We present the p-values for correlations in Fig. 5 in the text; see our response to editor's remark "L184-188".

We added to the caption of Fig. 6: Both correlations are highly significant (p < 0.002), indicating strong non-random associations between variables.

We added to the caption of Fig. 7: Panels b-e show highly significant correlations (p < 0.002), while panel f demonstrates no significant relationship (p = 0.35).

L257: Suggest rephrasing the subsection title as "4.1 Model performance and limitations"

We followed the editor's advice.

L279-285: The statements in this paragraph should clearly distinguish between findings derived from the model and those from observations or literature.

All presented findings in this paragraph represent model results unless observations are explicitly noted. We have added clarifications to distinguish between simulated and empirical data.

We start this paragraph now: Model simulations show that near-bottom oxygen deficiency in the Oder Lagoon occurs as widespread, episodic events. The most affected …

L285: Please delete the citation of Fig. 1. Fig. 1 shows the location of Oder River but can't see the high nutrient loads. Citing Fig. 1 is misleading here.

We deleted the reference.

L290-291: The claim that the newly introduced limnic phytoplankton group was "necessary to achieve realistic biomass concentrations" requires substantiation. Please clarify: 1) where are the results for this group's abundance and its impact on biomass presented (e.g., which figure)? 2) what evidence supports the conclusion of "necessity"? Was this based on a model sensitivity experiment that compared simulations with and without this functional group? 3) please moderate the language to reflect the specific context of this study. Does this finding imply that this group is universally necessary for all model setups in this region, or was it a required component within the specific configuration of this model?

We have introduced a table showing the fraction of the model phytoplankton groups in the Oder Lagoon. Furthermore, we clarified the raised questions by rewriting the whole paragraph. We think, the role of the limnic phytoplankton group is now much better described. We also emphasis the fact, that our findings could be useful for model configurations of the whole Baltic Sea. The revised text:

The newly introduced phytoplankton functional group (limnic phytoplankton) is by far the most abundant model phytoplankton group (Tab. 2). This new group was necessary to achieve realistic biomass concentrations in the lagoon. The limnic group's adaptation to low-light, CDOM-rich conditions enables realistic phytoplankton biomass simulation in the model. In environments outside the lagoon, where salinity levels are substantially higher, the limnic phytoplankton group becomes effectively absent from the community composition. This exclusion results from growth limitations imposed by elevated salinity conditions, which exceed the group's threshold. This mechanism, in combination with the other three groups, allows us to apply the biogeochemical model ERGOM in coastal waters of the Baltic Sea as well as in the open Baltic Sea without parameter tuning. This is especially important when the model is set up for the entire Baltic Sea at a high spatial resolution, for example, 2~km or finer, where lagoons are partly resolved. In this case, the ecosystem model provides reasonable results in coastal waters, lagoons, and the open Baltic Sea. We applied this Baltic Sea model to create open boundary conditions for the Oder Lagoon model, ensuring proper domain connectivity with the larger Baltic Sea model. Thus, simulations with a coarse-grained model deliver nearly seamless data for the open boundaries of the local model setups.

L292-293: The statement that the model can be applied in "coastal waters as well as in the open Baltic Sea" is somewhat vague. Please specify if "coastal waters" refers specifically to the Baltic Sea's coastal areas or to coastal waters in a global, general sense.

We clarified this imprecise formulation in the revised paragraph provided in our response to the previous editor's comment.

L294: Please express the spatial resolution in kilometers instead of nautical miles.

Changed into 2 km.

L300: The reference to Appendix A is too general. Please provide a brief summary regarding

which aspects of model performance are analyzed in the appendix and state the overall conclusion of that evaluation to give the reader necessary context.

We have given the validation summary and reference to Appendix A already in chapter 3.1. Thus, we decided to delete this sentence because we think it is superfluous.

L305-312: Please clarify the methodological basis of the three cited studies. E.g., observational or modelling.

We added information on the used methods; budget calculations based on observations and observed removal rates.

L310-311: Please cite the specific figure to support this statement.

We refer to Table 1 which show the numbers.

L330: Please specify the exact panel (e.g., Fig. 5b) and report the corresponding correlation coefficient and p-value.

We corrected the sentence: A significant correlation (p < 0.001) could be established for this relationship (Fig. 5b).

L337: Please specify the spatial resolution that is considered necessary (e.g., "on the order of 1 km or finer") to resolve the key coastal filter processes discussed in this study.

We added (on the order of 1~km or finer) to the sentence.

L357-359: The concluding statement that "For phosphorus, we did not find a similar dependence" on water residence time appears to be inconsistent with the visual data presented in Fig. 6b, which suggests a strong positive correlation. This claim cannot be definitively supported without a statistical test. Please clarify.

We appreciate the editor's careful review. As correctly indicated by the p-values in Fig. 6, phosphorus retention shows a significant relationship with residence time, and we have revised the text accordingly to reflect this finding.

Additionally, both nitrogen and phosphorus retention show significant dependence on water residence time in the lagoon system.